# WAVELET-BASED POSITIONAL REPRESENTATION FOR LONG CONTEXT

**Yui Oka, Taku Hasegawa, Kyosuke Nishida, Kuniko Saito**
NTT Human Informatics Laboratories, NTT Corporation
`yui.oka@ntt.com`

## ABSTRACT

In the realm of large-scale language models, a significant challenge arises when extrapolating sequences beyond the maximum allowable length. This is because the model's position embedding mechanisms are limited to positions encountered during training, thus preventing effective representation of positions in longer sequences. We analyzed conventional position encoding methods for long contexts and found the following characteristics. (1) When the representation dimension is regarded as the time axis, Rotary Position Embedding (RoPE) can be interpreted as a restricted wavelet transform using Haar-like wavelets. However, because it uses only a fixed scale parameter, it does not fully exploit the advantages of wavelet transforms, which capture the fine movements of non-stationary signals using multiple scales (window sizes). This limitation could explain why RoPE performs poorly in extrapolation. (2) Previous research as well as our own analysis indicates that Attention with Linear Biases (ALiBi) functions similarly to windowed attention, using windows of varying sizes. However, it has limitations in capturing deep dependencies because it restricts the receptive field of the model. From these insights, we propose a new position representation method that captures multiple scales (i.e., window sizes) by leveraging wavelet transforms without limiting the model's attention field. Experimental results show that this new method improves the performance of the model in both short and long contexts. In particular, our method allows extrapolation of position information without limiting the model's attention field.

## 1 INTRODUCTION

Several pre-trained large language models based on Transformer architecture (Vaswani et al., 2017) have demonstrated robust capabilities in various generative tasks (Devlin et al., 2019; Raffel et al., 2020; Brown et al., 2020; Touvron et al., 2023a; Jiang et al., 2023). However, limitations on the input sequence length arise due to the computational resource constraints encountered during the pre-training phase. Such constraints necessitate a determination of the maximum allowable length of sequences, hereinafter $L_{\text{train}}$, prior to the pre-training process, thus hindering the model's performance in processing sequences longer than those encountered during training. This weakness is primarily attributed to the positional encoding's ineffectiveness in handling sequences that exceed the length of those encountered during the model's training phase (Devlin et al., 2019; Press et al., 2022).

Rotary Position Embedding (RoPE) (Su et al., 2021) has become a common approach in many language models that handle long contexts, and it employs a rotation matrix to encode positional information and facilitate the processing of long sequences. To manage sequences longer than those encountered during training, various scaling strategies (Chen et al., 2023; bloc97, 2023; Peng et al., 2024; Liu et al., 2024) have been applied to RoPE, although these often require additional fine-tuning and incur further learning costs in addition to those of pre-training. In contrast, Attention with Linear Biases (ALiBi) (Press et al., 2022) is able to sequence length estimation beyond the limits of pre-training without requiring additional fine-tuning. However, ALiBi limits the attention's receptive field (Chi et al., 2023) in the manner of windowed attention (Beltagy et al., 2020). For this reason, a model using ALiBi may not be able to obtain information that is in a distant dependency relationship. In this paper, we analyze conventional positional encoding methods for long contexts,

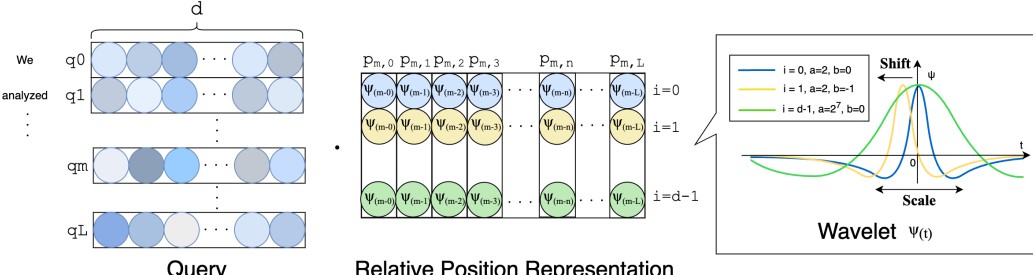

Figure 1: Overview of Wavelet-based Relative Positional Representation As in RPE (Shaw et al., 2018), our method computes a relative positional representation $(p_{m,n})^T$ to the query $q_m$ and the key $k_n$. Instead of learnable embedding in RPE, the position is computed based on the wavelet function. Different wavelet functions $\psi_{a,b}$ are used for each dimension of the head $d$. Furthermore, the scale parameter $a$ and the shift parameter $b$ change depending on the dimension of the head $d$.

and we propose a novel positional representation that permits extrapolation without constraining the attention mechanism's receptive field. First, we mathematically show that RoPE performs a process similar to a wavelet transformation—considered the gold standard of time-frequency analysis methodology. We interpreted the position of each token in the sequence as a time point in time-frequency analysis. However, RoPE does not perform a transformation in accordance with the order of positions but rather in accordance with the number of dimensions, and it does not capture the dynamic change in a signal over time. Furthermore, the values corresponding to the wavelet scale (i.e., window size) are constant, so RoPE does not make good use of the key characteristic of wavelet transforms, which is the ability to analyze signals on multiple scales. In other words, RoPE may fail to capture the dynamic change in a signal over time, such as what occurs in natural language. In this study, we also show that ALiBi provides different window sizes for each head.

Based on these insights, we propose a wavelet transform-based method, using multiple window sizes, to offer a robust and flexible approach to positional encoding. By performing a wavelet transform along the order of positions and introducing various scale parameters, our method can capture the dynamic changes in a sequence over positions in the manner of the original feature of wavelet transformation, i.e., time-frequency analysis. Following the methodology of Relative Position Representation (RPE) (Shaw et al., 2018), we implement our method with relative ease.

From our experiments on extrapolation capabilities using the wikitext-103 dataset (Merity et al., 2017), the results demonstrate that our method surpasses traditional positional encoding methods in perplexity. We also report that our method has lower perplexity than RoPE in experiments with long contexts using the Llama-2 model (Touvron et al., 2023b) and the CodeParrot dataset.

## 2 BACKGROUND

### 2.1 POSITIONAL REPRESENTATION

Within the Transformer architecture, positional encoding is employed to accurately represent the sequential position of each token. Positional encoding can be divided into two main types: absolute position, which expresses the position of a token from the static beginning of the sequence, and relative position, which expresses the position of each token in relation to the other tokens within the sequence. RoPE (Su et al., 2021), which adopts a type of absolute position, uses a rotation matrix to compute the position and then multiplies it by the query and key to represent the position. RPE (Shaw et al., 2018), based on a type of relative position, uses a learnable embedding that represents the position of distances of up to 16 or 32 tokens by clipping. Two other variations include T5 Bias (Raffel et al., 2020), which has an enlarged RPE window size, and Transformer-XL (Dai et al., 2019), which uses a sine wave for position representation instead of learnable embedding.

Position encoding plays a critical role in enabling models to effectively handle long context sequences, and it allows for extrapolation. Relative position is not a position expression that depends on the length of the sequence, so it is effective in extrapolation. ALiBi (Press et al., 2022) is an

effective position representation method for extrapolation: It uses the relative position bias of all tokens by adding a linear bias to each head's attention score, rather than using position embedding. However, ALiBi is unable to obtain information in a distant dependency relationship due to its constraints on the self-attention mechanism's receptive field(Chi et al., 2023). On the other hand, absolute position is unsuitable for extrapolation because it expresses the position of all words in the sequence. For this reason, many methods have been proposed for fine-tuning RoPE by interpolating positions in using absolute position (Chen et al., 2023; bloc97, 2023; Peng et al., 2024).

## 2.2 FREQUENCY ANALYSIS AND TIME-FREQUENCY ANALYSIS

Frequency analysis in signal processing involves analyzing the frequency components of a signal to understand its behavior. The Fourier transform (FT) (Bracewell & Bracewell, 1986) is a key method for frequency analysis, converting a signal from the time domain to the frequency domain, thus providing a global view of its frequency content. However, the FT does not provide any information about **when** specific frequencies occur. To address this limitation, time-frequency analysis techniques have been applied. The wavelet transform (WT) (Grossmann & Morlet, 1984; Mallat, 1989) offers a more flexible approach by analyzing the signal at multiple scales or resolutions. The WT adaptively provides high time resolution for high-frequency components and high frequency resolution for low-frequency components, making it well-suited for analyzing signals with non-stationary or transient features. This adaptability allows the wavelet transform to capture both time and frequency information with varying degrees of precision.

## 3 ROPE AND WAVELET TRANSFORM

### 3.1 PRELIMINARY

**Wavelet Transform**   A wavelet is a wave that decays quickly and locally as it approaches zero. A function $\psi$ defined on a real $\mathbb{R}$ is called a wavelet function if it belongs to the space $L^2(R)$ of square integrable functions and satisfies the following conditions:

$$\int_{-\infty}^{\infty} \mid \psi(x) \mid^2 dx < \infty. \tag{1}$$

The wavelet function is defined as follows.

$$\psi_{a,b}(t) = \frac{1}{\sqrt{a}}\psi\Big(\frac{t-b}{a}\Big). \tag{2}$$

In this case, $b$ is the shift and $a > 0$ is the scale parameter. The scale parameter $a$ simultaneously changes the range over which the wavelet is localized as well as the wavelet's amplitude. Typical wavelets include the Haar wavelet (Haar, 1910), Ricker wavelet (Ricker, 1944), and Morlet wavelet (Bernardino & Santos-Victor, 2005). Suppose that we sample $T$ values at regular intervals from a continuous signal. Wavelet transform (WT) (Grossmann & Morlet, 1984) is the process of transforming a signal $x(t)$ into the frequency domain and time domain by computing the inner product of the wavelet function $\psi_{a,b}(t)$ and signal $x(t)$.

$$W(a,b) = \sum_{t=0}^{T-1} \psi_{a,b}(t)x(t). \tag{3}$$

In some cases, the term "Discrete Wavelet Transform" or "Wavelet Transform" is used to refer to multi-resolution analysis (Mallat, 1989), but in this paper we follow the original definition. We can see that the FT only converts to the frequency domain, whereas the WT converts to two domains: scale $a$ and shift $b$. For example, consider the case of a conversion to two scales and four shifts. When $a \in [2, 4]$ and $b \in [0, 1, 2, 3]$, the wavelet transform can be expressed in terms of determinants as follows:

$$\begin{bmatrix} W(2,0) \\ W(4,0) \\ W(2,1) \\ W(4,1) \\ \vdots \\ W(4,3) \end{bmatrix} = \begin{bmatrix} \psi_{2,0}(0) & \psi_{2,0}(1) & \psi_{2,0}(2) & ... & \psi_{2,0}(T-1) \\ \psi_{4,0}(0) & \psi_{4,0}(1) & \psi_{4,0}(2) & ... & \psi_{4,0}(T-1) \\ \psi_{2,1}(0) & \psi_{2,1}(1) & \psi_{2,1}(2) & ... & \psi_{2,1}(T-1) \\ \psi_{4,1}(0) & \psi_{4,1}(1) & \psi_{4,1}(2) & ... & \psi_{4,1}(T-1) \\ \vdots & \vdots & \vdots & \ddots & \vdots \\ \psi_{4,3}(0) & \psi_{4,3}(1) & \psi_{4,3}(2) & ... & \psi_{4,3}(T-1) \end{bmatrix} \begin{bmatrix} x(0) \\ x(1) \\ x(2) \\ \vdots \\ x(T-1) \end{bmatrix}. \tag{4}$$

Furthermore, since $\psi_{a,b}(t) = \psi_{a,0}(t-b)$ from Eq.2, $\psi$ of the wavelet transform in Eq 4 is expressed as follows.

$$
\begin{bmatrix} W(2,0) \\ W(4,0) \\ W(2,1) \\ W(4,1) \\ \vdots \\ W(4,3) \end{bmatrix} = \begin{bmatrix} \psi_{2,0}(0) & \psi_{2,0}(1) & \psi_{2,0}(2) & \dots & \psi_{2,0}(T-1) \\ \psi_{4,0}(0) & \psi_{4,0}(1) & \psi_{4,0}(2) & \dots & \psi_{4,0}(T-1) \\ \psi_{2,0}(-1) & \psi_{2,0}(0) & \psi_{2,0}(1) & \dots & \psi_{2,0}(T-2) \\ \psi_{4,0}(-1) & \psi_{4,0}(0) & \psi_{4,0}(1) & \dots & \psi_{4,0}(T-2) \\ \vdots & \vdots & \vdots & \ddots & \vdots \\ \psi_{4,0}(-3) & \psi_{4,0}(-2) & \psi_{4,0}(-1) & \dots & \psi_{4,0}(T-3) \end{bmatrix} \begin{bmatrix} x(0) \\ x(1) \\ x(2) \\ \vdots \\ x(T-1) \end{bmatrix}. \tag{5}
$$

Due to the characteristics of the scale parameter $a$, the values of the wavelet matrix become 0 or approach 0 outside a certain range that depends on the specific wavelet function.

**RoPE** RoPE incorporates positional information directly into the self-attention mechanism by rotating the query and key vectors in complex space. When divided into even and odd dimensions, the following calculations are performed for the $m$-th query in each sequence. In even dimensions, RoPE is expressed as follows.

$$
\begin{bmatrix} q_0^m \\ q_2^m \\ \vdots \\ q_{d-2}^m \end{bmatrix} = \begin{bmatrix} \cos m\theta_1 & -\sin m\theta_1 & 0 & 0 & \dots & 0 & 0 \\ 0 & 0 & \cos m\theta_2 & -\sin m\theta_2 & \dots & 0 & 0 \\ \vdots & \vdots & \vdots & \vdots & \ddots & \vdots & \vdots \\ 0 & 0 & 0 & 0 & \dots & \cos m\theta_{d/2} & -\sin m\theta_{d/2} \end{bmatrix} \begin{bmatrix} q_0^m \\ q_1^m \\ \vdots \\ q_{d-2}^m \\ q_{d-1}^m \end{bmatrix}. \tag{6}
$$

where $q^m \in \mathbb{R}^{1 \times d}$ is the $m$-th query when the number of dimensions is $d$ and $\theta_i = 10000^{-2(i-1)/d}, i \in [1, 2, ..., d/2]$. For RoPE in odd dimensions, see Appendix A.1. The same process is also performed for the $n$-th key $k^n \in \mathbb{R}^{1 \times d}$.

## 3.2 THEORETICAL ANALYSIS

First, we show the wavelet transform using the following two Haar-like wavelets (Haar, 1910).

$$
\psi(t) = \begin{cases} \cos f(t) & 0 \le t < 1, \\ -\sin f(t) & 1 \le t < 2, \\ 0 & \text{otherwise}. \end{cases} \quad \psi'(t) = \begin{cases} \sin f(t) & 0 \le t < 1, \\ \cos f(t) & 1 \le t < 2, \\ 0 & \text{otherwise}. \end{cases} \tag{7}
$$

Here, $f : \mathbb{R} \to \mathbb{R}$ is a function that satisfies $\int_{-\infty}^{\infty} \psi(t)\, dt = 0$ and Eq.(1). Assuming that when $x(t)(0 \le t \le d-1)$ is a signal with $d$ elements, the wavelet $\psi$ is used and wavelet transform is performed at each scale $a = 1$. We define the shift parameter as $b_j = j - \delta(j)(j = 0, 2, .., d-2)$. Here, $\delta(t)$ is a function such that $0 \le t \le d-1$ and $0 \le \delta(t) < 1$. When the wavelet function is Haar-like wavelet $\psi(t)$ in Eq.(7) and $a = 1$ and $b \in [b_0, b_2, .., d_{d-2}]$, the wavelet matrix $\psi$ in the wavelet transform $w = \psi x$ can be expressed in terms of determinants as follows.

$$
\begin{bmatrix} W(1, b_0) \\ W(1, b_2) \\ \vdots \\ W(1, b_{d-2}) \end{bmatrix} = \begin{bmatrix} \cos\phi_0 & -\sin\phi_1 & 0 & 0 & \dots & 0 & 0 \\ 0 & 0 & \cos\phi_2 & -\sin\phi_3 & \dots & 0 & 0 \\ \vdots & \vdots & \vdots & \ddots & \vdots & \vdots & \vdots \\ 0 & 0 & 0 & 0 & \dots & \cos\phi_{d-2} & -\sin\phi_{d-1} \end{bmatrix} \begin{bmatrix} x(0) \\ x(1) \\ \vdots \\ x(d-2) \\ x(d-1) \end{bmatrix}. \tag{8}
$$

To simplify the notation in the matrix representation above, we write $\phi_j$ for $j = 0, 1, \ldots, d-1$, where $\phi_j = f(1 + \delta(j))$ if $j$ is odd, and $\phi_j = f(\delta(j))$ otherwise. Let $x$ be the query $q^m$, and define $f$ such that $\phi_j = \phi_{j+1} = m\theta_{\lceil \frac{j+1}{2} \rceil}$ for $j = 0, 2, 4, \ldots, d-2$, where $\theta_i = 10000^{-2(i-1)/d}$ and $i \in [1, 2, ..., d/2]$. Under this definition, the transformation matrix of Eq. (8) becomes identical to that of Eq. (6) in RoPE. [1] In other words, RoPE can be viewed as a wavelet transform using Haar-like wavelets that change amplitude on a fixed scale. Furthermore, the same result as RoPE in odd

---

[1] The proof of the existence of $f(t)$ that satisfies this condition is provided in Appendix A.2.

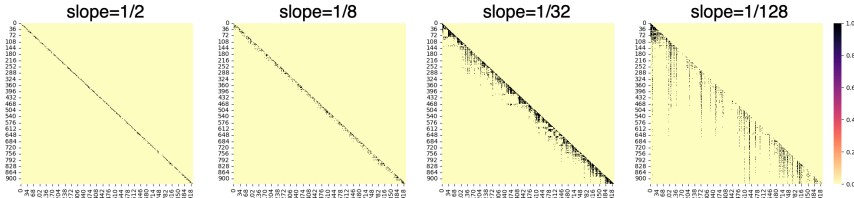

Figure 2: Heatmap of scaled attention scores via softmax normalization in ALiBi without non-overlapping inference. The vertical axis represents the query, while the horizontal axis corresponds to the key in the attention map. For clarity, values of 0.001 or more are mapped to black, while values below that are mapped to yellow. The maximum allowable length of sequences is $L_{\text{train}} = 512$, and the inference length is 1012.

dimensions can be obtained when using $\psi'$ for wavelet transformation. [2] This wavelet transform in RoPE is performed across the number of query head dimensions $d$. Therefore, RoPE can be considered a wavelet transformation along the head dimension using a wavelet with a fixed scale of 2.[3]

## 4 Window Size Variability in ALiBi

ALiBi has a restricted receptive field and behaves in the manner of windowed attention (Chi et al., 2023; Beltagy et al., 2020). A receptive field refers to the specific region of the input space that significantly influences the model's output, typically representing the area where the most relevant features are captured. ALiBi is expressed as

$$\text{softmax}(q_m K^T + slope \cdot [-(m-1), \ldots, -2, -1, 0]), \tag{9}$$

where the $slope$ is a head-specific slope fixed before training and $K^T \in \mathbb{R}^{m \times d}$ is the first $m$ keys. In this section, we analyzed the window size in ALiBi using the attention map.

### 4.1 Insights from Attention Map Analysis

A heatmap of scaled attention scores obtained through softmax normalization is shown in Figure 2. The number of heads $N$ is 8, and the slope of ALiBi is $\left[\frac{1}{2}, \frac{1}{4}, \frac{1}{8}, \frac{1}{16}, \frac{1}{32}, \frac{1}{64}, \frac{1}{128}, \frac{1}{256}\right]$. In extrapolation, sequences are often divided, but in this section the sequences are not divided. The experimental setting was set to the same as that in Section 6.1. The perplexity results are shown in Table 1.

The attention map shows that ALiBi uses multiple window sizes corresponding to relative positions and that the window size increases as the slope decreases. Moreover, previous research (Chi et al., 2023) shows that constraining the window size (slope) to a single value leads to increased perplexity. Consequently, one of the reasons ALiBi is effective, compared to a previous relative position using fixed window sizes in T5 Bias (Raffel et al., 2020), is its ability to accommodate multiple window sizes. ALiBi does not perform calculations like those in Eq. (3), so it does not exactly match the wavelet transform. However, having windows of various sizes is similar to the role of the scale parameter used in wavelet transforms.

## 5 Wavelet-based Positional Representation

Wavelet transform (WT) is a method of analyzing signals using variable-scale wavelets, and it is possible to adjust the scale of the window. This scalability allows both broad and fine signal features to be efficiently extracted by shifting the wavelet while changing the window size. In particular, this is suitable for investigating non-stationary signals. For this reason, we believe that the wavelet

---

[2]Additionally, when $\sin m\theta_i = \cos m\theta_i$, the Haar wavelet matrix and RoPE are the same when the scale is 2, and the shift is $[2, 4, \ldots, d/2]$. Refer to Appendix A.3 for the detailed proof.

[3]From previous research(Tancik et al., 2020), we also hypothesized that this could be equivalent to a Fourier transform. However, this hypothesis does not hold (refer to Appendix A.4 for details).

transform approach is effective for capturing the dynamic fluctuations of signals that change over time, and it is also effective for the fluid nature of natural language, which is not constrained by periodicity. Furthermore, when extrapolating, it is important to be able to respond flexibly to changes in context and information. For this reason, we believe that the wavelet transform is also an effective method for extrapolation.

When applying wavelet transforms to positional encoding, a key question arises: Which features should be leveraged for handling long-context dependencies? Notably, RoPE shares conceptual similarities with the wavelet transform (Section 3); however, RoPE depends on absolute positional information, which limits its effective context window to the training length ($L_{\text{train}}$) and restricts its extrapolation capabilities. In contrast, ALiBi offers extrapolation capabilities by using relative position, and it supports varying window sizes (Section 4). However, ALiBi's linear bias constrains its receptive field, making it insufficient for capturing long-range dependencies. According to Press et al. (2022), conventional relative positional encoding (RPE) methods (Shaw et al., 2018; Raffel et al., 2020), which rely on a fixed window size, are similarly ineffective for extrapolation. In conclusion, we adopt relative position with flexible window sizes to handle long-context and extrapolation.

Accordingly, we propose positional representation based on wavelet transform with the following characteristics:

1. **Position-based Transformation**: RoPE predominantly relies on independent transformation based on the 'head' dimensions. ALiBi employs multiple windows based on the relative position of the sentence, rather than the dimension of the head, which may contribute to its performance. Therefore, we apply a wavelet transform based on the relative position of the sentence.

2. **Type of Wavelet**: RoPE can be thought of as a wavelet transform using the Haar wavelet, which is the simplest wavelet. However, Haar wavelets might fall short in capturing the intricacies of natural languages. Transitioning toward the use of more sophisticated wavelet functions could enhance our approach to distilling and representing a broader spectrum of features inherent in natural languages.

3. **Diversification of Window Sizes (Scale Parameters)**: From our analysis of ALiBi, we found that having multiple windows is effective for long contexts. The original version of RoPE works with a single fixed scale. To address this limitation, we introduce a variety of scale and shift parameters.

### 5.1 METHODOLOGY

**Incorporating Wavelet Transform into PE** Due to the wavelet shift feature, we adopt relative position representation using ALiBi because it is more suitable than absolute position representation. [4] In a transformer model (Vaswani et al., 2017), the self-attention mechanism operates by projecting the input sequence into three distinct representations—queries ($Q$), keys ($K$), and values ($V$)—using learnable weight matrices. Self-attention sublayers employ $N$ attention heads. In self-attention sublayers, $e_{m,n}$ is the attention score for each query, and then the key is calculated. RPE(Shaw et al., 2018) expresses position by calculating the inner product of the query and the relative position embedding. We incorporate the wavelet function into RPE as follows.

$$e_{m,n} = \frac{q_m k_n^T + q_m (p_{m,n})^T}{\sqrt{d}}, \tag{10}$$

where $q_m$ is the $m$th query ($q_m \in \mathbb{R}^{1 \times d}, 1 \leq m \leq L$) of a sentence of length $L$, $k_n$ is the $n$th key ($k_n \in \mathbb{R}^{1 \times d}, 1 \leq n \leq L$) for $q_m$, and $d$ is the number of dimensions of each head. Here, $p_{m,n}$ is the relative position from the $m$-th query to the $n$-th key. RPE (Shaw et al., 2018) uses learnable embedding for $p_{m,n} \in \mathbb{R}^d$ and a fixed scale by clipping. However, instead of using learnable embeddings to represent $p_{m,n}$, we use $d$-pattern wavelet functions with multiple scales to calculate the position. In our method, there is no clipping, and the distance of the position expression is fixed regardless of the length of the sentence.

---

[4]We also considered incorporating wavelet transforms into RoPE, but decided not to do this because it would make the computational cost even higher. A discussion on this is included in Appendix A.5.

**Wavelet Function**    In conventional wavelets, such as in Eq. (2), the amplitude also varies depending on the scale parameter $a$. In the proposed method, all amplitudes are the same.

$$\psi_{a,b}(t) = \psi\Big(\frac{t-b}{a}\Big). \tag{11}$$

The variable $t$ is assigned the relative position, which is $t = m - n$. We used the Ricker wavelet (Ricker, 1944) as a base wavelet, which is formulated as follows.

$$\psi(t) = (1 - t^2)\exp\Big(\frac{-t^2}{2}\Big). \tag{12}$$

**Shift and scale parameters**    We use $s$ distinct patterns for the scale parameter $a$ and $\frac{d}{s}$ patterns for the shift parameter $b$.

$$(a, b) \in \{2^0, 2^1, 2^2, ...2^{s-1}\} \times \{0, 1, 2, 3, ..., \frac{d}{s} - 1\}. \tag{13}$$

The scale parameter is a power of 2 derived from the principles of the discrete wavelet transform. By combining the $\frac{d}{s}$-pattern shift parameters $b$ with the $s$-pattern scale parameters $a$, we generate $d$ distinct wavelets. In this way, our method can set the $s$-pattern **context window size** using the scale parameter $a$ and the $d$-pattern **context window** using both the scale parameter $a$ and the shift parameter $b$. For instance, with a head dimension of $d = 128$, we use $s = 8$ scale variants ($a \in \{2^0, 2^1, ..., 2^7\}$) and 16 shift variants ($b \in \{0, 1, 2, ..., 15\}$), resulting in $8 \times 16 = 128$ unique wavelets. Finally, $p_{m,n}$ is computed as follows.[5]

$$p_{m,n} = \Big(1 - \Big(\frac{m-n-b}{a}\Big)^2\Big)\exp\Big(-\frac{1}{2}\Big(\frac{m-n-b}{a}\Big)^2\Big). \tag{14}$$

# 6    SHORT-CONTEXT EXPERIMENT

## 6.1    EXPERIMENTAL SETTINGS

First, we conducted a small-scale experiment to compare our approach with various position encodings. We used the WikiText-103 dataset (Merity et al., 2017), which consists of over 103 million tokens of English Wikipedia articles. We performed a comparative evaluation using a Transformer-based language model (Baevski & Auli, 2019). The dimensionality of the word embedding $d_{model}$ is 1024, the number of heads $N$ is 8, the dimensionality of the heads $d$ is 128, and the number of layers is 16. The implementation was based on the fairseq (Ott et al., 2019)-based code[6] provided in a previous work(Press et al., 2022), and all hyperparameters were set to the same values as those in the literature(Press et al., 2022).[7] The maximum allowable lengths of sequences were set to $L_{\text{train}} = 512$ and $L_{\text{train}} = 1024$.

**Compared Methods**    Although $\theta = 10,000$ is usually used for RoPE, it has been found that extending $\theta$ to $500,000$ is effective for long contexts (Xiong et al., 2024). Therefore, we compared $\theta = 10,000$ with $\theta = 500,000$. In addition to ALiBi and RoPE, the following position representations were also compared: NoPE (Kazemnejad et al., 2023), in which position information is given, and TransXL (Dai et al., 2019), which is a relative positional representation that uses sine waves.

**Evaluation Metric**    We use perplexity as our evaluation metric. Following previous research (Press et al., 2022), we evaluated the validation set. To evaluate sequences longer than $L_{\text{train}}$ tokens, it is common to divide the sequence into $L_{\text{train}}$-length sub-sequences, evaluate each independently, and report the average score. However, methods that use relative positions to express a wide range, such as ALiBi, Trans-Xl, and the proposed method, are able to consider a wider range of contexts than $L_{\text{train}}$. For this reason, in this paper, we report not only the perplexity of non-overlapping inference but also the normal perplexity when the sequence is not divided into partial sequences. Note

---

[5]Implementation tips for reducing the memory and computational efficiency of the proposed method are included in Appendix A.6.

[6]https://github.com/ofirpress/attention_with_linear_biases

[7]See Appendix A.7 for more details of hyperparameters.

Table 1: Perplexity of validation set in extrapolation experiments using Wikitext-103. Maximum allowable lengths of sequences in pre-training are $L_{\text{train}} = 512$ and $L_{\text{train}} = 1024$.

| | | Sequence Length | | | | | | | | | |
| --- | --- | --- | --- | --- | --- | --- | --- | --- | --- | --- | --- |
| | | $L_{\text{train}} = 512$ | | | | | | $L_{\text{train}} = 1024$ | | | |
| | pos | 128 | 256 | 512 | 1012 | 1512 | 2512 | 1024 | 1524 | 3024 | 5024 |
| Perplexity in Non-overlapping Inference with $L_{\text{train}}$ | | | | | | | | | | | |
| NoPE(Kazemnejad et al., 2023) | - | 26.38 | 23.23 | 21.53 | 21.52 | 21.53 | 21.53 | 20.81 | 21.52 | 21.49 | 21.45 |
| RoPE (Su et al., 2021) | abs | 23.82 | 20.98 | 19.39 | 19.35 | 19.39 | 19.38 | 18.42 | 19.51 | 19.52 | 19.48 |
| RoPE (Xiong et al., 2024) | abs | 23.81 | 20.95 | 19.35 | 19.32 | 19.35 | 19.33 | 18.50 | 19.53 | 19.54 | 19.50 |
| Trans-XL (Dai et al., 2019) | rel | 24.16 | 21.53 | 19.96 | 19.92 | 19.93 | 19.96 | 18.67 | 19.75 | 19.74 | 19.70 |
| ALiBi(Press et al., 2022) | rel | 24.18 | 21.32 | 19.69 | 19.64 | 19.69 | 19.64 | 18.66 | 19.64 | 19.65 | 19.62 |
| Wavelet(Ricker) | rel | **23.64** | **20.82** | **19.19** | **19.15** | **19.17** | **19.20** | **18.26** | **19.30** | **19.34** | **19.26** |
| Perplexity without Non-overlapping Inference | | | | | | | | | | | |
| NoPE(Kazemnejad et al., 2023) | - | 26.38 | 23.23 | 21.53 | 21.03 | 21.58 | 48.48 | 20.81 | 20.45 | 22.11 | 59.37 |
| RoPE (Su et al., 2021) | abs | 23.82 | 20.98 | 19.39 | 23.25 | 44.38 | 93.94 | 18.42 | 18.29 | 33.20 | 122.52 |
| RoPE (Xiong et al., 2024) | abs | 23.81 | 20.95 | 19.35 | 23.70 | 40.39 | 77.90 | 18.50 | 18.30 | 29.25 | 83.43 |
| Trans-XL(Dai et al., 2019) | rel | 24.16 | 21.53 | 19.96 | 19.09 | 18.92 | 19.05 | 18.67 | 18.25 | 18.17 | 18.76 |
| ALiBi(Press et al., 2022) | rel | 24.18 | 21.32 | 19.69 | 18.71 | 18.42 | 18.41 | 18.66 | 18.14 | 17.86 | 17.88 |
| Wavelet(Ricker) | rel | **23.64** | **20.82** | **19.19** | **18.23** | **18.00** | **17.99** | **18.26** | **17.13** | **17.14** | **17.44** |
| Haar (Fixed scale) | rel | 24.98 | 22.07 | 20.49 | 51.61 | 116.87 | 299.26 | - | - | - | - |
| Haar | rel | 23.73 | 20.89 | 19.27 | 18.34 | 18.11 | 18.17 | - | - | - | - |
| Morlet | rel | 24.15 | 21.28 | 19.65 | 19.02 | 20.46 | 26.56 | - | - | - | - |
| Gaussian | rel | 23.77 | 20.90 | 19.30 | 18.31 | 18.02 | 17.88 | - | - | - | - |

that when the sequence length is less than $L_{\text{train}}$, the scores for the perplexity of non-overlapping inference and the normal perplexity without division into partial sequences are the same. Of course, when perplexity is considered without division into partial sequences, the performance of RoPE is expected to decrease greatly because unknown values are used for RoPE when processing a sequence longer than the length encountered during training.

## 6.2 MAIN RESULTS

The experimental results are shown in Table 1. The results of perplexity in inference without overlap show that the proposed method using wavelets achieved the lowest perplexity and was also effective for extrapolation. In RoPE, the values used during training are also used in inference without overlap, so the perplexity remains low even when the sequence length exceeds $L_{\text{train}}$. At the same time, however, perplexity is higher for ALiBi and Trans-XL than for RoPE, which is attributed to the limited context range of the position representation's applicability due to the division of the sequence into sub-sequences. In contrast, the proposed method maintains low perplexity even in the case of division into sub-sequences, suggesting that the wavelet position representation is highly effective.

On the other hand, perplexity without non-overlapping inference showed the opposite results. First, since RoPE uses absolute positions, it is necessary to use new values for unknown positions, and thus perplexity increased significantly. However, in the case of $\theta = 500,000$, the increase in perplexity was relatively small. On the contrary, Trans-XL and ALiBi, which use relative positions, were able to handle longer contexts, and perplexity decreased as the range of position representations expanded. In the proposed method, perplexity also decreased and the best score was achieved. Trans-XL uses a position representation based on a periodic sine wave function, but the proposed method, which uses wavelets, could further decrease perplexity. This result supports our claim (section 5) that an approach like wavelet transformation is more effective than periodic functions in capturing the fluid nature of natural language, which is not constrained by periodicity.

## 6.3 ANALYSIS

### 6.3.1 HOW EFFECTIVE ARE THE OTHER WAVELET TYPES?

We also conducted experiments to see whether the same effect could be obtained with other wavelets. The wavelets tested were the Gaussian-based wavelet $\psi(t) = exp(-t^2)$, the Morlet-based wavelet $\psi(t) = exp(-t^2)cos(at)$, and the Haar-based wavelet. Note that when $\psi(t/a)$ exists in our Morlet wavelet, the frequency of this cosine wavelet is not affected by the scale parameter $a$.

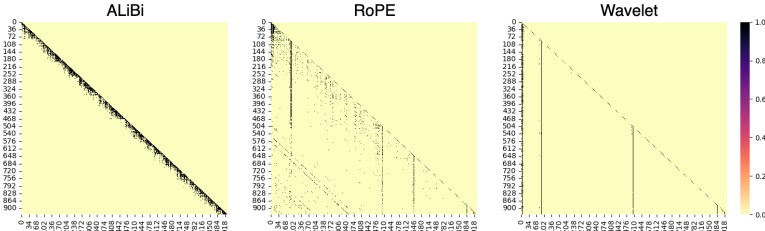

Figure 3: Heatmap of scaled attention scores via softmax normalization in 4th head after softmax operation without non-overlapping inference. The vertical axis represents the query, while the horizontal axis corresponds to the key. For clarity, values of 0.001 or more are mapped to black, while values below that are mapped to yellow. The maximum allowable length of sequences in pre-training is $L_{\text{train}} = 512$ and the inference length is 1012. See Appendix A.12 for other heads.

We used the following formula for the Haar wavelet.

$$
\psi(t) = \begin{cases} 1 & -0.5 \le t < 0, \\ -1 & -1 \le t < -0.5, \\ 0 & \text{otherwise.} \end{cases} \tag{15}
$$

We kept the shift and scale parameters constant, only changing the wavelet function. We also tested the Haar wavelet when set to $a \in \{2^0, 2^0, 2^0, ...2^0\}$. Consequently, this restricted Haar wavelet had the same scale parameter setting as the RoPE demonstrated in Section 3.2. [8] The graphs of these wavelet functions are shown in Appendix A.10 (Fig. 6). Extrapolation experiments were conducted under the same conditions as the experimental setup in Section 6, with $L_{\text{train}} = 512$ during training.

As shown in Table 1, the Ricker-, Haar- and Gaussian-based wavelets had lower perplexity than the Morlet wavelet. One possibility is that complex wavelets with multiplied cosine waves, such as Morlet wavelets, are not suitable for relative positional representation. On the other hand, wavelets with all positive values, such as Gaussian-based wavelets, are expected to represent positions within a narrower distance than the window specified by the scale parameter due to softmax normalization. This suggests that wavelets with a specific range of negative values are suitable, like a Ricker wavelet, for positional representation. Although the Haar wavelet is simple, it is such a wavelet with negative values within a specific range. Therefore, it is considered effective, although not as much as a Ricker wavelet. However, when the scale parameter is restricted ( $a \in \{2^0, ..., 2^0\}$), as in RoPE, the perplexity increases. This demonstrates the importance of having multiple scales, or in this case, window sizes. We also performed ablation studies for each shift and scale parameter (Appendix A.13) and for discrete wavelets as well as continuous wavelets (Appendix A.14).

### 6.3.2 CAN IT HANDLE TOKENS WITH LONG-RANGE DEPENDENCIES?

Figure 3 shows the attention map of scaled attention scores obtained through softmax normalization for the proposed method. The inference length is $L = 1012$ without non-overlapping inference. The most notable feature of the proposed method is that it is always able to attend to specific tokens. The words that always receive attention are those that are important in the sentence, such as the special token, the first token, and the subject of the sequence. On the other hand, ALiBi has a restricted receptive field for attention, making it unable to capture long-distance dependencies. Similar to the proposed method, RoPE emphasizes important and special words but struggles to capture those that are farther apart. Moreover, as the sentence lengthens, it loses the ability to attend to the initial word. This tendency was also seen in sentences shorter than $L_{\text{train}}$. Accordingly, the proposed method has demonstrated its superiority at capturing long dependencies without restricting the receptive field of attention.

---

[8]Normally, the wave is localized when $t > 0$ in the Haar wavelet, but in the decoder model, only the range $t < 0$ is used. Therefore, we transformed the Haar wavelet into a form that reflects the original function f(x) across the y-axis.

Table 2: Perplexity in Non-overlapping Inference with $L_{\text{train}} = 4096$.

|  | Sequence Length | | | |
|---|---|---|---|---|
|  | 4 k | 8 k | 16 k | 32 k |
| RoPE (Xiong et al., 2024) | 9.45 | 9.33 | 9.12 | 8.90 |
| Wavelet | 9.00 | 9.01 | 8.83 | 8.60 |

# 7 LONG CONTEXT

## 7.1 EXPERIMENTAL SETTINGS

Next, we conducted a large-scale experiment using a Llama-based model (Touvron et al., 2023b). We pre-trained the Llama-2-7B[9] model from scratch. For pre-training, we used the RedPajama dataset (Computer, 2023), which selects a 1B-token sample of all samples. The maximum allowable length of sequences in pre-training was set to $L_{\text{train}} = 4096$. For the same reason as given in Section 6.1, we set $\theta = 500,000$ for RoPE. Furthermore, when the scale parameter is $a \in \{2^0, 2^1, ..., 2^7\}$, the range within which the wavelet is localized becomes narrow. Therefore, in our method, we changed the scale parameter to $a \in \{2^2, 2^3, ..., 2^9\}$. The other parameters are the same as those used for the Llama-2-7B model(Touvron et al., 2023b). We used CodeParrot [10] for evaluation, which is good for long-distance testing because it requires an understanding of patterns and contextualization of information over long distances. [11]

## 7.2 MAIN RESULTS

The experimental results are shown in Table 2. Regardless of whether interpolation or extrapolation was applied, the perplexity of our method was lower than RoPE. Therefore, even with large-scale models and long contexts, our method was found to be effective. Moreover, the results in Section 6.2 show that not dividing the sequence further reduces perplexity. Therefore, our method might also be able to further reduce perplexity. We investigated the use of LongBench(Bai et al., 2024), with the results given in Appendix A.15.

In addition, position interpolation methods (Chen et al., 2023; bloc97, 2023; Peng et al., 2024; Ding et al., 2024) have been proposed to adapt RoPE for longer contexts. We believe these methods can be integrated into our approach for the following reasons. First, the parameter $\theta$ in RoPE corresponds to the scale parameter $a$ in our method, implying compatibility between the two frameworks. Both $\theta$ and $a$ refer to the upper limit of the number of positions to be expressed. Second, the LongRoPE paper (Ding et al., 2024) reveals that performance improves when extrapolation is avoided for the initial positions, which likely aligns with the shift parameter $b$ in our method. Thus, it is highly likely that existing position interpolation methods will integrate seamlessly with our approach.

# 8 CONCLUSION

In this paper, we demonstrated that RoPE can be interpreted as a wavelet transform, and we introduced a novel positional representation method that leverages the wavelet transform's advantages, effectively capturing positional information across various window sizes. Our experimental results demonstrate the proposed method's superior performance in extrapolation tasks when compared to traditional positional representation techniques. Importantly, our approach offers the advantage of not constraining the receptive field, which allows more flexible and comprehensive analysis of positions. Calculating relative positions is known to require more resources than calculating absolute positions, so we show methods for reducing memory consumption in Appendix A.6. However, the computational overhead of calculating relative positions may still impose a bottleneck, and thus reducing it is an important direction for future work.

---

[9] https://huggingface.co/meta-llama/Llama-2-7b
[10] https://huggingface.co/datasets/codeparrot/codeparrot-clean
[11] See Appendix A.8 for more details of the hyperparameters.

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

# A APPENDIX

## A.1 ROTARY POSITION EMBEDDING

RoPE incorporates positional information directly into the self-attention mechanism by rotating the query and key vectors in the complex space. When divided into even and odd dimensions, the following calculations are performed for the $m$-th query in each sequence. In even dimensions, RoPE is expressed as follows.

$$
\begin{bmatrix} q_0^m \\ q_2^m \\ \vdots \\ q_{d-2}^m \end{bmatrix} = \begin{bmatrix} \cos m\theta_1 & -\sin m\theta_1 & 0 & 0 & \dots & 0 & 0 \\ 0 & 0 & \cos m\theta_2 & -\sin m\theta_2 & \dots & 0 & 0 \\ \vdots & \vdots & \vdots & \vdots & \ddots & \vdots & \vdots \\ 0 & 0 & 0 & 0 & \dots & \cos m\theta_{d/2} & -\sin m\theta_{d/2} \end{bmatrix} \begin{bmatrix} q_0^m \\ q_1^m \\ \vdots \\ q_{d-2}^m \\ q_{d-1}^m \end{bmatrix}.
$$
(16)

In odds dimensions, RoPE is expressed as follows.

$$
\begin{bmatrix} q_1^m \\ q_3^m \\ \vdots \\ q_{d-1}^m \end{bmatrix} = \begin{bmatrix} sin\theta_1 & cos\theta_1 & 0 & 0 & \dots & 0 & 0 \\ 0 & 0 & sinm\theta_2 & cosm\theta_2 & \dots & 0 & 0 \\ \vdots & \vdots & \vdots & \vdots & \ddots & \vdots & \vdots \\ 0 & 0 & 0 & 0 & \dots & sinm\theta_{d/2} & cosm\theta_{d/2} \end{bmatrix} \begin{bmatrix} q_0^m \\ q_1^m \\ \vdots \\ q_{d-2}^m \\ q_{d-1}^m \end{bmatrix},
$$
(17)

where $q^m \in \mathbb{R}^{1 \times d}$ is the $m$-th query when the number of dimensions is $d$ and $\theta_i = 10000^{-2(i-1)/d}, i \in [1, 2, ..., d/2]$. The same process is also performed for the $n$-th key $k^n \in \mathbb{R}^{1 \times d}$.

## A.2 PROOF OF THE EXISTENCE OF $f(t)$

We prove the existence of $f(t)$ as described in 3.2 such that $\phi_j = \phi_{j+1} = m\theta_{\lceil \frac{j+1}{2} \rceil}$, where $\theta_i = 10000^{-2(i-1)/d}$ and $i \in [1, 2, ..., d/2]$. Here, we restrict our proof to $\psi(t)$ in Eq.(7), but a similar argument can be applied to $\psi'(t)$, following analogous steps to establish its validity.

First, we revisit the definition of $\psi(t)$:

$$
\psi(t) = \begin{cases} \cos f(t) & 0 \le t < 1, \\ -\sin f(t) & 1 \le t < 2, \\ 0 & \text{otherwise.} \end{cases}
$$
(18)

Here, $f : \mathbb{R} \to \mathbb{R}$ is a monotonous function that satisfies $\int_{-\infty}^{\infty} \psi(t)\, dt = 0$ and Eq.(1). Assuming that when $x(t)(0 \le t \le d - 1)$ is a signal with $d$ elements, the wavelet $\psi$ is used and wavelet transform is performed at each scale $a = 1$. We define the shift parameter as $b_j = j - \delta(j)(j = 0, 2, .., d - 2)$. Here, $\delta(t)$ is a monotonous function such that $0 \le t \le d - 1$ and $0 \le \delta(t) < 1$.

$$
\begin{bmatrix} W(1, b_0) \\ W(1, b_2) \\ \vdots \\ W(1, b_j) \\ \vdots \\ W(1, b_{d-2}) \end{bmatrix} = \begin{bmatrix} \cos \phi_0 & -\sin \phi_1 & 0 & 0 & \dots & \dots & 0 & 0 \\ 0 & 0 & \cos \phi_2 & -\sin \phi_3 & \dots & \dots & 0 & 0 \\ \vdots & \vdots & \vdots & \ddots & \vdots & \vdots & \vdots & \vdots \\ 0 & 0 & 0 & \dots & \cos \phi_j & -\sin \phi_{j+1} & \dots & 0 \\ \vdots & \vdots & \vdots & \vdots & \vdots & \ddots & \vdots & \vdots \\ 0 & 0 & 0 & 0 & \dots & \dots & \cos \phi_{d-2} & -\sin \phi_{d-1} \end{bmatrix} \begin{bmatrix} x(0) \\ x(1) \\ \vdots \\ x(j) \\ x(j+1) \\ \vdots \\ x(d-2) \\ x(d-1) \end{bmatrix}.
$$
(19)

To simplify the notation in the matrix representation above, we write $\phi_j$ for $j = 0, 1, \dots, d - 1$, where $\phi_j = f(1 + \delta(j))$ if $j$ is odd, and $\phi_j = f(\delta(j))$ otherwise. We let $x$ be the query $q^m$. The function $f(t)$ is defined such that $0 < f(t) \le 2k\pi$ for $0 \le t < 1$ and $0 < f(t) \le 2k\pi$ for $1 \le t < 2$, where $k$ is the smallest natural number satisfying $m < 2k\pi$.

**Do Haar-like wavelets satisfy the necessary conditions of a wavelet?** Here, $f(t)$ must be a function such that $\psi(t)$ satisfies the conditions of a wavelet. For Eq. 1, it is evident that it holds for any $f$ satisfying $0 < f(t) \leq 2k\pi$. Next, we consider the zero-mean property. As an example, consider $f(t)$ defined as $f(t) = 2k\pi t$ for $0 \leq t < 1$ and $f(t) = 2k\pi(t-1)$ for $1 \leq t < 2$. If we set $\theta = f(t)$, we have:

$$\int_{-\infty}^{\infty} \psi(t)dt = \int_0^{2k\pi} \cos\theta d\theta + \int_0^{2k\pi} -\sin\theta d\theta = 0. \tag{20}$$

Since this satisfies the zero-mean property, we conclude that there exists an $f(t)$ such that $\psi(t)$ is a wavelet.

Furthermore, we observe that there exists a $\delta(t)$ satisfying $\phi_j (= f(\delta(j))) = \phi_{j+1}(= f(1+\delta(j))) = 2k\pi\delta(t) = m\theta_{\lceil \frac{j+1}{2} \rceil}$ for $j = 0, 2, \ldots, d-2$. In other words, we can simply choose a function $\delta(j)$ that satisfies $\delta(j) = \dfrac{m\theta_{\lceil \frac{j+1}{2} \rceil}}{2k\pi}$ for $j = 0, 2, \ldots, d-2$.

### A.3 HAAR WAVELET

Here, we explain wavelet transform using the Haar wavelet, which is the simplest wavelet. The definition of the Haar wavelet is as follows.

$$\psi(t) = \begin{cases} 1 & 0 \leq t < 1/2, \\ -1 & 1/2 \leq t < 1, \\ 0 & otherwise. \end{cases} \quad \phi(t) = \begin{cases} 1 & 0 \leq t < 1, \\ 0 & otherwise. \end{cases} \tag{21}$$

Haar wavelets are defined not only by a wavelet function $\psi$ but also by a scaling function $\phi$.

The method of analyzing signals by performing a discrete wavelet transform using these two functions is called multi-resolution analysis. When the scale is fixed at 2 and the shift $b \in [0, 2, ..., d/2]$, the wavelet transform using the wavelet function and scaling function is expressed as follows.

$$\begin{bmatrix} \psi_{2,0}(0) & \psi_{2,0}(1) & \psi_{2,0}(2) & \psi_{2,0}(3) & ... & \psi_{2,0}(T-2) & \psi_{2,0}(T-1) \\ \phi_{2,0}(0) & \phi_{2,0}(1) & \phi_{2,0}(2) & \phi_{2,0}(3) & ... & \phi_{2,0}(T-2) & \phi_{2,0}(T-1) \\ \psi_{2,0}(-2) & \psi_{2,0}(-1) & \psi_{2,0}(0) & \psi_{2,0}(1) & ... & \psi_{2,0}(T-4) & \psi_{2,0}(T-3) \\ \phi_{2,0}(-2) & \phi_{2,0}(-1) & \phi_{2,0}(0) & \phi_{2,0}(1) & ... & \phi_{2,0}(T-4) & \phi_{2,0}(T-3) \\ \vdots & \vdots & \vdots & \vdots & ... & \vdots & \vdots \\ \psi_{2,0}(-\frac{d}{2}) & \psi_{2,0}(-\frac{d}{2}+1) & \psi_{2,0}(-\frac{d}{2}+2) & \psi_{2,0}(-\frac{d}{2}+3) & ... & \psi_{2,0}(0) & \psi_{2,0}(1) \\ \phi_{2,0}(-\frac{d}{2}) & \phi_{2,0}(-\frac{d}{2}+1) & \phi_{2,0}(-\frac{d}{2}+2) & \phi_{2,0}(-\frac{d}{2}+3) & ... & \phi_{2,0}(0) & \phi_{2,0}(1) \end{bmatrix} \begin{bmatrix} x(0) \\ x(1) \\ x(2) \\ \vdots \\ x(T-2) \\ x(T-1) \end{bmatrix}. \tag{22}$$

From Eq.(21), $\psi_{2,0}$ and $\phi_{2,0}$ are as follows.

$$\psi_{2,0}(t) = \begin{cases} 1/\sqrt{2} & 0 \leq t < 1, \\ -1/\sqrt{2} & 1 \leq t < 2, \\ 0 & otherwise. \end{cases} \quad \phi_{2,0}(t) = \begin{cases} 1/\sqrt{2} & 0 \leq t < 2, \\ 0 & otherwise. \end{cases} \tag{23}$$

Therefore, the Haar wavelet transform is a $2 \times 2$ block matrix.

$$\begin{bmatrix} \psi(2,0) \\ \phi(2,0) \\ \psi(2,2) \\ \phi(2,2) \\ \vdots \\ \psi(2,T-2) \\ \phi(2,T-2) \end{bmatrix} = \begin{bmatrix} 1/\sqrt{2} & -1/\sqrt{2} & 0 & 0 & ... & 0 & 0 \\ 1/\sqrt{2} & 1/\sqrt{2} & 0 & 0 & ... & 0 & 0 \\ 0 & 0 & 1/\sqrt{2} & -1/\sqrt{2} & ... & 0 & 0 \\ 0 & 0 & 1/\sqrt{2} & 1/\sqrt{2} & ... & 0 & 0 \\ \vdots & \vdots & \vdots & \vdots & ... & \vdots & \vdots \\ 0 & 0 & 0 & 0 & ... & 1/\sqrt{2} & -1/\sqrt{2} \\ 0 & 0 & 0 & 0 & ... & 1/\sqrt{2} & 1/\sqrt{2} \end{bmatrix} \begin{bmatrix} x(0) \\ x(1) \\ x(2) \\ x(3) \\ \vdots \\ x(T-2) \\ x(T-1) \end{bmatrix}. \tag{24}$$

This matrix is the Haar forward transform using matrix multiplication for a $T$ element signal. This matches the RoPE matrix with $m\theta = \pi/4$.

## A.4 ISN'T RoPE A FOURIER TRANSFORM?

We also hypothesized that this could be equivalent to a Fourier transform. However, this hypothesis does not hold. When a signal $x(t)$ that changes over time is Fourier transformed, its spectrum $F(k)$ is obtained. The process of converting an actual discrete signal $x(t)$ into a spectrum $F(k)$ is as follows.

$$F(f) = \sum_{t=0}^{T} x(t) w^{f \cdot t}. \tag{25}$$

The Fourier transform can be expressed as a matrix formula as follows.

$$
\begin{bmatrix} F(0) \\ F(1) \\ F(2) \\ \vdots \\ F(f) \end{bmatrix} =
\begin{bmatrix}
w^{0 \cdot 0} & w^{0 \cdot 1} & w^{0 \cdot 2} & \dots & w^{0 \cdot (T-1)} \\
w^{1 \cdot 0} & w^{1 \cdot 1} & w^{1 \cdot 2} & \dots & w^{1 \cdot (T-1)} \\
w^{2 \cdot 0} & w^{2 \cdot 1} & w^{2 \cdot 2} & \dots & w^{2 \cdot (T-1)} \\
\vdots & \vdots & \vdots & \ddots & \vdots \\
w^{f \cdot 0} & w^{f \cdot 1} & w^{f \cdot 2} & \dots & w^{f \cdot (T-1)}
\end{bmatrix}
\begin{bmatrix} x(0) \\ x(1) \\ x(2) \\ \vdots \\ x(T-1) \end{bmatrix}. \tag{26}
$$

Here, $f \in \mathbb{R}$ is the wave number, $T \in \mathbb{R}$ is the number of samples, and $i$ is the imaginary unit. $w = exp(-\frac{2\pi i}{T})$ is called the Twiddle Factor (Gentleman & Sande, 1966), which is a complex number expressed in polar form using Euler's formula $e^{-i\theta} = cos\theta - isin\theta$. In the complex plane, $w^{f \cdot t}$ represents a point on the unit circle with an argument of the complex number $-\frac{ft2\pi}{T}$. From this formula, we can see that the Fourier transform calculates the inner product of all signals and sine waves. However, in RoPE, the inner product with sine waves is calculated only within each block.

Next, when calculating the attention score with RoPE, does the Fourier transform hold? Attention scores of the $m$-th query $q^m$ and the $n$-th key $k^n$ with RoPE are calculated as follows.

$$
\left[ R_m^1 (Q_m^1)^T, ..., R_m^{d/2} (Q_m^{d/2})^T \right]
\begin{bmatrix} R_n^1 K_n^1 \\ \vdots \\ R_n^{d/2} K_n^{d/2} \end{bmatrix}
= \sum_{i=1}^{d/2} (Q_m^i)^T R_{n-m}^i K_n^i, \tag{27}
$$

where $Q_m^{d/2}$ is the query divided into every two dimensions, and $R_m^{d/2}$ is the rotation matrix.

$$
Q_m^{d/2} = \begin{bmatrix} q_m^{d-1} \\ q_m^d \end{bmatrix}, K_n^{d/2} = \begin{bmatrix} k_n^{d-1} \\ k_n^d \end{bmatrix}, R_m^{d/2} = \begin{bmatrix} cosm\theta_{d/2} & -sinm\theta_{d/2} \\ sinm\theta_{d/2} & cosm\theta_{d/2} \end{bmatrix}.
$$

Aligning with the Fourier transform, as illustrated in Equation 26, requires a process involving the inner product between a frequency tensor of dimensions f×T and a signal tensor of dimensions T×1 (such as the query vector). However, RoPE operates on independent 2×2 blocks, where each block is processed separately. Consequently, RoPE's block-wise operations do not conform to the structure required by the Fourier transform. Moreover, if we focus solely on the RoPE and key operations in Equation 27, they may appear to align with the structure of a Fourier transform. However, since the final step involves taking the inner product with the query, the overall operation deviates from the path of becoming a perfect match with the Fourier transform. Furthermore, the rotation factor represents a rotation in the complex plane, and even if it is expressed as in Eq.(26) using a rotation matrix, it does not completely match a rotation matrix that represents a rotation in the Euclidean plane.

Therefore, RoPE cannot be equated with the Fourier transform. Furthermore, even if it were the same as the Fourier transform, it would be unsuitable for processing non-stationary signals and thus unsuitable for processing natural language, which is a non-stationary flow.

## A.5 CONSIDERATION OF WAVELET TRANSFORMATION BASED ON RoPE

In this paper, we explore the incorporation of wavelet transforms into RoPE (Relative Positional Encoding) following our previous discussion on RPE (Relative Position Encoding). In this regard, integrating wavelet transforms into RoPE presents challenges for controlling computational and memory costs. In Sections 3 and 4, we highlighted the potential effectiveness of employing multiple scales for extrapolation. With this in mind, we present a simplified formula for applying various scales and wavelet transforms to RoPE, which we refer to here as a RoPE-based Wavelet.

$$
\begin{bmatrix} q_0^m \\ q_1^m \\ q_2^m \\ q_3^m \\ \vdots \\ q_{d-1}^m \\ q_{d-2}^m \end{bmatrix} = \begin{bmatrix} \cos m\theta_1 & -\sin m\theta_1 & 0 & 0 & ... & 0 & 0 \\ \sin m\theta_1 & \cos m\theta_1 & 0 & 0 & ... & 0 & 0 \\ \cos m\theta_2 & -\sin m\theta_2 & \cos m\theta_2 & -\sin m\theta_2 & ... & 0 & 0 \\ \sin m\theta_2 & \cos m\theta_2 & \sin m\theta_2 & \sin m\theta_2 & ... & 0 & 0 \\ \vdots & \vdots & \vdots & \vdots & \ddots & \vdots & \vdots \\ \cos m\theta_{d/2} & -\sin m\theta_{d/2} & \cos m\theta_{d/2} & -\sin m\theta_{d/2} & ... & \cos m\theta_{d/2} & -\sin m\theta_{d/2} \\ \sin m\theta_{d/2} & \cos m\theta_{d/2} & \sin m\theta_{d/2} & \sin m\theta_{d/2} & ... & \sin m\theta_{d/2} & \cos m\theta_{d/2} \end{bmatrix} \begin{bmatrix} q_0^m \\ q_1^m \\ q_2^m \\ q_3^m \\ \vdots \\ q_{d-2}^m \\ q_{d-1}^m \end{bmatrix},
$$

(28)

where $q^m \in \mathbb{R}^{1 \times d}$ is the $m$-th query when the number of dimensions is $d$ and $\theta_i = 10000^{-2(i-1)/d}, i \in [1, 2, ..., d/2]$. The same process is also performed for the $n$-th key $k^n \in \mathbb{R}^{1 \times d}$.

Conversely, the method introduced in Section 5 is here called RPE-based Wavelet. The key differences between RoPE-based Wavelet and RPE-based Wavelet are as follows:

- **Number of Scale Parameters**: In RPE-based Wavelet, the scale parameters can be selected up to the maximum sequence length. However, in RoPE-based Wavelet, the selection is limited to a maximum of $d$.

- **Memory Usage**: RoPE-based Wavelet requires a wavelet matrix that corresponds to the number of absolute positions $m$. Consequently, the memory usage is significantly higher. Unlike RoPE-based Wavelet, RPE-based Wavelet does not necessitate a wavelet matrix that matches $m$ values, allowing the use of Tip 2 from Appendix A.6, which improves memory efficiency.

- **Absolute and Relative Positions**: When applying wavelet transforms using RoPE-based Wavelet, it is necessary to use absolute positions. In contrast, RPE-based Wavelet can use relative positions, which enhances extrapolation.

- **Computational Cost**: Implementing wavelet transforms via RoPE-based Wavelet requires processing both the query and the key, necessitating two calculations. RPE-based Wavelet, as discussed in Section 5, only requires one computation, since it processes only the query.

Additionally, we conducted an experiment with RoPE-based Wavelet. **Unfortunately, we had to halt the learning process because it took over five times longer than anticipated.** Considering the learning costs associated with large-scale language models in recent years, we believe the RoPE-based Wavelet approach is not feasible.

## A.6 Implementation Tips for Wavelet Position Representation

**Tip 1**   Similar to RPE(Shaw et al., 2018), we used Eq. (10) as

$$\alpha_{ij} = softmax\Big(\frac{q_i K^T + q_i(p_{ij})^T}{\sqrt{d_k}}\Big).$$

By transforming it in this way, it is possible to reduce the computational complexity to $O(batch \times n \times length^2 \times d + length^2 \times d)$, where $batch$ is the batch size, $n$ is the number of heads, $length$ is the number of tokens, and $d$ is the number of dimensions of each head. The experiments in Section 6 are implemented based on the methodology introduced in this section.

**Tip 2**   When dealing with long contexts of over 4 k with a large model, the memory efficiency of $(d, length, length)$ of the wavelet position becomes a bottleneck. Therefore, we further reduce the memory usage to $(d, length)$ by using `torch.scatter` to scatter the wavelet position representation to the attention mask. In the relative position representation in the decoder, only the position information of the token before the current token is required, for example, $0, -1, -2, etc.$ Therefore, we pre-compute the information up to $0, -1, -2, ...length$ and reduce the memory usage by using `torch.scatter` to distribute it. Specifically, we prepare a $(d, length)$ wavelet tensor and calculate the 2D inner product with the query, which has been transposed to $(length \times batch, d)$. The tensor after the calculation becomes $(length \times batch, length)$, which is then scattered using `torch.scatter` so that it becomes a relative position in the attention mask. This reduces the amount of memory used from $(d, length, length)$ to $(d, length)$, and the calculation can be performed using calculations between 2D tensors. The experiments in Section 7 are implemented based on the methodology introduced in this section.

## A.7 Experimental Settings in Short-Context Experiment

The parameter settings used in the extrapolation experiments were the same as those in the original ALiBi paper. The dimensionality of the word embedding $d_{model}$ is 1024, the number of heads $N$ is 8, the dimensionality of the heads $d$ is 128, and the number of layers is 16. The implementation was based on the fairseq (Ott et al., 2019)-based code[12] provided in a previous work(Press et al., 2022), and all hyperparameters were set to the same values as those in the literature(Press et al., 2022). The number of training epochs is 205, and the batch size is 9216. The learning rate was set to 1.0, and the learning process was updated by 1e-7 every 16,000 steps.

## A.8 Experimental Settings in Long-Context Experiment

The dimensionality of the word embedding $d_{model}$ is 4096, the number of heads $N$ is 32, the dimensionality of the heads $d$ is 128, and the number of layers is 32. The number of training steps is 30,000, and the batch size is 1. The learning rate was set to 0.0003. We used AdamW(Loshchilov & Hutter, 2019) as the optimizer, with $(\beta_1, \beta_2) = (0.9, 0.95)$. In accordance with previous research (Rubin & Berant, 2024; Wu et al., 2022; Zhang et al., 2024), we then used 100 sampled sequences in the training set for evaluation. In this experiment, due to the large model size and long sequence length, we report perplexity only for non-overlapping inference using $L_{train}$, since the memory capacity is exceeded.

---

[12]`https://github.com/ofirpress/attention_with_linear_biases`

## A.9 RICKER WAVELET

Figures 4 and 5 show the Ricker wavelets with multiple scale $a$.

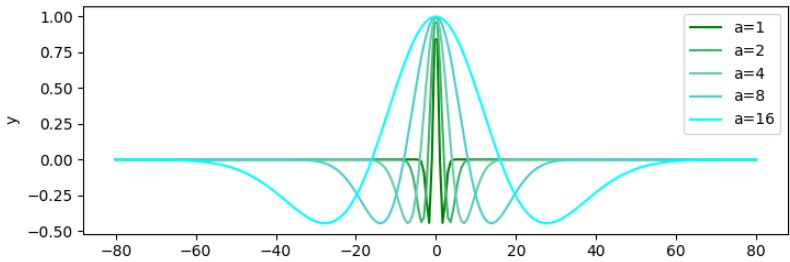

Figure 4: Graph of compared Ricker wavelet functions with $a = [2^0, 2^1, 2^2, 2^3, 2^4]$

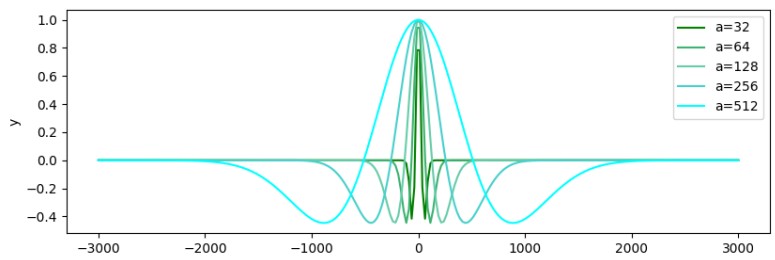

Figure 5: Graph of compared Ricker wavelet functions with $a = [2^5, 2^6, 2^7, 2^8, 2^9]$

## A.10 WAVELET TYPE

Figure 6 shows graphs of the wavelets compared in Section 6.3.1. It can be seen that the simplest is the Haar wavelet, while the most complex is the Morlet wavelet.

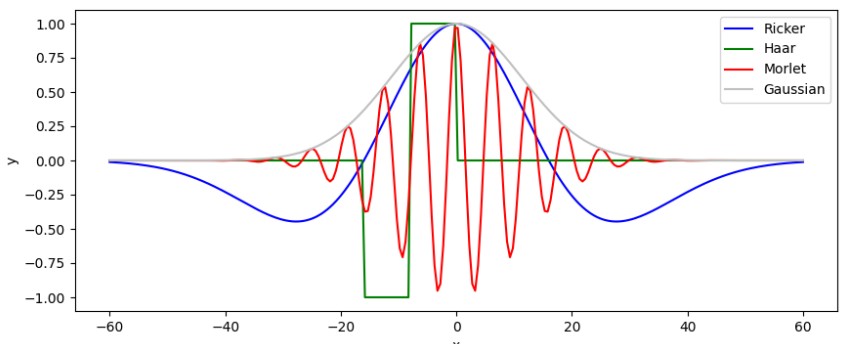

Figure 6: Graph of compared wavelet functions. The case with scale parameter $a = 2^4$ and shift parameter $b = 0$ is shown.

## A.11 EXAMPLE OF HEAT MAP AND TEXT CORRESPONDENCE

Figure 7 shows the attention map after softmax operation for the proposed method. First, the notable feature of the proposed method is that it is always able to pay attention to specific tokens. The words that always receive attention are those that are important in the sentence, such as the '' token, the first token, and words that are the subject of the sequence, such as 'he.' Moreover, as with ALiBi, the proposed method has a different scope of attention for each head.

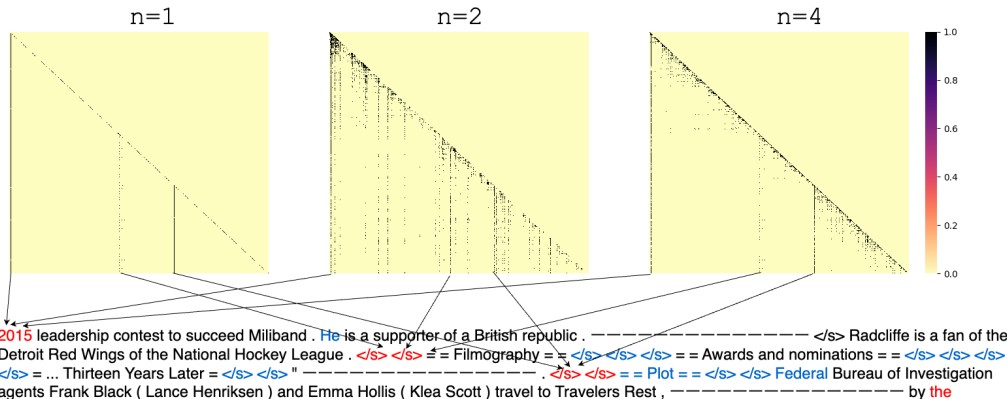

Figure 7: Heatmap of attention score $e_{ij}$ after softmax operation for the proposed method. The maximum sequence length is $L_{max} = 512$, and the sequence length at inference is $L = 1012$. From left to right, $n = 1, 2, 4$th heads are shown. Scores above 0.01 are mapped in black and the rest in yellow. Words that were always given attention in all heads are shown in red, and words that were frequently given attention only in the $n = 2$nd head are shown in blue. Sentences are omitted in the middle because they are long with 1012 tokens.

## A.12 CAN IT HANDLE TOKENS WITH LONG-RANGE DEPENDENCIES?

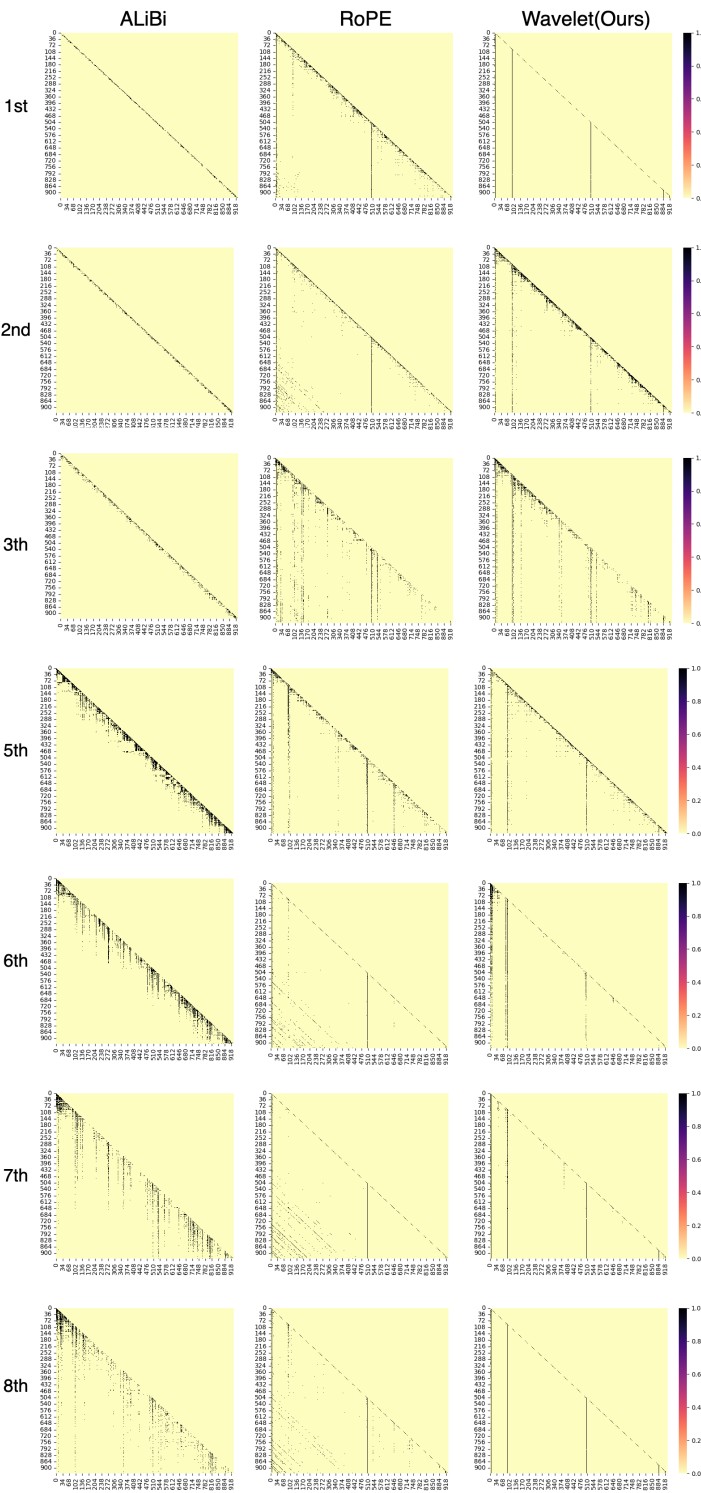

Figure 8: Heatmap of scaled attention scores via softmax normalization in 1-3rd and 5-8th head after softmax operation for ALiBi, RoPE, and our method. For clarity, values of 0.001 or more are mapped to black, while values below that are mapped to yellow.

Table 3: Perplexity of validation set in extrapolation experiments using Wikitext-103. Maximum allowable length of sequences in pre-training is $L_{\text{train}} = 512$.

| | | | Sequence Length | | | | | |
|---|---|---|---|---|---|---|---|---|
| | scale $a$ | shift $b$ | 128 | 256 | 512 | 1012 | 1512 | 2512 |
| *Perplexity without Non-overlapping Inference* | | | | | | | | |
| Ricker | $\{2^0, 2^1, ..., 2^7\}$ | $\{0, 1, 2, ..., 15\}$ | 23.64 | 20.82 | 19.19 | 18.23 | 18.00 | 17.99 |
| Ricker | $\{2^1, 2^2 ..., 2^8\}$ | $\{0, 1, 2, ..., 15\}$ | 23.77 | 20.89 | 19.25 | 18.23 | 17.97 | 18.02 |
| Ricker | $\{2^2, 2^3 ..., 2^9\}$ | $\{0, 1, 2, ..., 15\}$ | 23.92 | 21.03 | 19.40 | 18.41 | 18.14 | 18.07 |
| Ricker | $\{2^0, 2^1, 2^2, 2^3\}$ | $\{0, 1, 2, ..., 31\}$ | 23.96 | 21.13 | 19.55 | 18.87 | 19.40 | 21.73 |
| Ricker | $\{2^0, 2^1\}$ | $\{0, 1, 2, ..., 63\}$ | 24.49 | 21.60 | 19.95 | 20.90 | 32.01 | 70.80 |
| Ricker | $\{2^0, 2^1 ..., 2^{15}\}$ | $\{0, 1, 2, ..., 7\}$ | 23.74 | 20.88 | 19.24 | 18.22 | 17.96 | 17.84 |
| Ricker | $\{2^0, 2^1 ..., 2^{31}\}$ | $\{0, 1, 2, 3\}$ | 23.75 | 20.86 | 19.26 | 18.24 | 17.96 | 17.84 |
| Ricker | $\{2^0, 2^1 ..., 2^{63}\}$ | $\{0, 1\}$ | 23.75 | 20.88 | 19.30 | 18.31 | 18.04 | 18.02 |
| Ricker | $\{2^0, 2^1 ..., 2^{127}\}$ | $\{0\}$ | 23.97 | 21.10 | 19.46 | 18.50 | 18.27 | 18.29 |
| Ricker | $\{2^7\}$ | $\{0, 1, 2, ..., 127\}$ | 24.35 | 21.45 | 19.80 | 20.68 | 20.87 | 21.31 |
| Gaussian | $\{2^0, 2^1, ..., 2^7\}$ | $\{0, 1, 2, ..., 15\}$ | 23.77 | 20.90 | 19.30 | 18.31 | 18.02 | 17.88 |
| Gaussian | $\{2^1, 2^2 ..., 2^8\}$ | $\{0, 1, 2, ..., 15\}$ | 23.92 | 21.02 | 19.41 | 18.41 | 18.15 | 18.01 |
| Gaussian | $\{2^2, 2^3 ..., 2^9\}$ | $\{0, 1, 2, ..., 15\}$ | 23.98 | 21.09 | 19.46 | 18.43 | 18.13 | 17.93 |
| Gaussian | $\{2^0, 2^1, 2^2, 2^3\}$ | $\{0, 1, 2, ..., 31\}$ | 23.83 | 29.96 | 19.33 | 18.43 | 18.40 | 18.94 |
| Gaussian | $\{2^0, 2^1\}$ | $\{0, 1, 2, ..., 63\}$ | 24.28 | 21.35 | 19.70 | 18.96 | 19.63 | 23.14 |
| Gaussian | $\{2^0, 2^1 ..., 2^{15}\}$ | $\{0, 1, 2, ..., 7\}$ | 23.72 | 20.86 | 19.24 | 18.24 | 17.95 | 17.77 |
| Gaussian | $\{2^0, 2^1 ..., 2^{31}\}$ | $\{0, 1, 2, 3\}$ | 23.78 | 20.92 | 19.29 | 18.30 | 18.01 | 17.85 |
| Gaussian | $\{2^0, 2^1 ..., 2^{63}\}$ | $\{0, 1\}$ | 23.86 | 20.98 | 19.37 | 18.46 | 18.20 | 18.10 |
| Gaussian | $\{2^0, 2^1 ..., 2^{127}\}$ | $\{0\}$ | 24.21 | 21.31 | 19.68 | 18.71 | 18.45 | 18.45 |
| Gaussian | $\{2^7\}$ | $\{0, 1, 2, ..., 127\}$ | 24.48 | 21.62 | 20.05 | 19.53 | 22.63 | 35.23 |
| Haar | - | - | 24.98 | 22.07 | 20.49 | 51.61 | 116.87 | 299.26 |
| Haar | $\{2^0, 2^1, ..., 2^7\}$ | $\{0, 1, 2, ..., 15\}$ | 23.73 | 20.89 | 19.27 | 18.34 | 18.11 | 18.17 |
| Morlet | $\{2^0, 2^1, ..., 2^7\}$ | $\{0, 1, 2, ..., 15\}$ | 24.15 | 21.28 | 19.65 | 19.02 | 20.46 | 26.56 |

## A.13 ABLATION STUDY OF SCALE AND SHIFT PARAMETERS

In this section, we present the findings from our ablation study focusing on the shift and scale parameters of the Ricker and Gaussian wavelets. As indicated in Table 1, both wavelet types demonstrate substantial effectiveness in our method. To further evaluate their performance, we explored the contributions of two parameters, i.e., the scale parameter $a$ and the shift parameter $b$, while keeping all other settings consistent with those outlined in Section 6.

**Results** The results of our experiments are summarized in Table 3. Both the Ricker and Gaussian wavelets exhibit similar trends regarding the influence of the scale and shift parameters on extrapolation performance. Initially, we observed that increasing the scale parameter value $a$ while holding the shift parameter $b$ ($\{2^0, 2^1, ..., 2^7\} \times \{0, 1, 2, ..., 15\}$, $\{2^1, 2^2, ..., 2^8\} \times \{0, 1, 2, ..., 15\}$ and $\{2^2, 2^3, ..., 2^9\} \times \{0, 1, 2, ..., 15\}$) constant maintained the performance of extrapolation, albeit with some fluctuations. Conversely, when we increased the number of shift parameters while decreasing the number of scale parameters ($\{2^0, 2^1, 2^2, 2^4\} \times \{0, 1, 2, ..., 31\}$ and $\{2^0, 2^1\} \times \{0, 1, 2, ..., 63\}$), there was a noticeable decline in performance. These findings underscore the significance of the scale parameters in extrapolation. Moreover, we found that increasing the number of scale parameters while decreasing the number of shift parameters led to performance improvements in some instances ($\{2^0, 2^1, ..., 2^{15}\} \times \{0, 1, 2, ..., 7\}$ and $\{2^0, 2^1, ..., 2^{31}\} \times \{0, 1, 2, 3\}$). However, when the shift parameters were reduced to two or entirely eliminated ($\{2^0, 2^1, ..., 2^{63}\} \times \{0, 1\}$ and $\{2^0, 2^1, ..., 2^{127}\}$), relying solely on the scale parameters resulted in a deterioration of extrapolation performance. Moreover, even when the scale parameter was fixed and only the shift parameter was used ($\{2^7\} \times \{0, 1, 2, .., 127\}$), the extrapolation performance decreased. This suggests the potential importance of shift parameters as well.

In conclusion, our analysis highlights the critical roles of both shift and scale parameters in the effectiveness of our wavelet-based method.

## A.14 Ablation Study of Wavelet types

In this section, we also explored a variety of wavelet types beyond those previously discussed. In Section 6.3.1, our focus was primarily on wavelets that could be computed directly from mathematical formulas. However, in this section, we expand our inquiry to include wavelets with varying numbers of vanishing moments as well as discrete wavelet transformations. Additionally, drawing from previous research (Wang et al., 2020), we considered the necessity for a distinct approach when incorporating complex numbers into positional encoding. Consequently, our study did not encompass wavelets that incorporate complex numbers.

**Wavelet types**  The specific wavelets under consideration in our investigation are outlined as follows:

- Daubechies (`db`) (Daubechies, 1992) - Compactly supported orthonormal wavelets
- Symlets (`sym`) - Wavelets with minimum asymmetry
- Coiflets (`coif`) - The scaling and wavelet functions have the same number of vanishing moments
- Meyer (`dmey`) - Wavelets defined in the frequency domain
- Biorthogonal Spline (`bior`) - Two wavelets are used: one for decomposition, and the other for reconstruction
- Reverse biorthogonal Spline (`rbio`)

In addition, the graphs of these wavelets are shown in Figures 9 and 10. As the number of vanishing moments increases, the wave oscillation becomes larger. Therefore, we also conducted a survey by vanishing point moment. The name of a wavelet is derived from the number of vanishing moments. For example, `db6` is a Daubechies wavelet with 6 vanishing moments, and `sym3` is a Symlet wavelet with 3 vanishing moments. In the case of Coiflet wavelets, `coif3` is a Coiflet wavelet with 6 vanishing moments. The names of `bior` and `rbio` wavelets are derived from the number of vanishing moments possessed by the decomposition and reconstruction wavelets, respectively. For example, `bior3.5` is Biorthogonal wavelet that has 3 vanishing moments for the decomposition wavelet and 5 vanishing moments for the reconstruction wavelet. Biorthogonal wavelets and Reverse-Biorthogonal wavelets can calculate the approximate values of decomposition wavelets and reconstruction wavelets, but in this case, we only used the values of decomposition wavelets.

**Experimental Settings**  We used Pywavelet (Lee et al., 2019) [13] to calculate the approximate values of these wavelets. In addition, in this experiment, we calculated the approximate values by specifying 8 levels of $\{1, 2, ..., 8\}$ instead of the 8-pattern scale parameters $\{2^0, 2^1, ..., 2^7\}$. We used the shift parameter $\{0, 1, 2, ..., 15\}$. The other experimental settings are the same as those in Section 6.

**Results**  The experimental results are summarized in Table 4. Overall, the performance observed was suboptimal for extrapolation. However, it is important to note that since the parameters were fixed at levels $\{1, 2, ..., 8\}$, we believe that performance may be enhanced with adjustments to these levels. Notably, the rbio1.1 wavelet demonstrated promising extrapolation capabilities, suggesting significant potential for future improvements. In contrast, the coif and dmey wavelets exhibited limited performance, even with shorter sequences, indicating their potential unsuitability for position encoding tasks. Conversely, while the extrapolation performance ($> 512$) of other wavelets was generally low, their interpolation performance ($\leq 512$) remained consistently stable, highlighting another avenue for enhancement. Furthermore, the performance of the db, bior, and rbio wavelets showed a positive correlation with an increasing number of vanishing points. This finding underscores the importance of vanishing points as a critical factor influencing performance. In conclusion, our analysis indicates that both the shape of the wavelet and the number of vanishing points play significant roles in determining extrapolation performance. Future work should explore these relationships further to identify optimal configurations for improved performance outcomes.

---

[13] https://pywavelets.readthedocs.io/en/latest/index.html

Table 4: Perplexity without Non-overlapping Inference. We evaluated the validation set in extrapolation experiments using Wikitext-103. The maximum allowable length of sequences in pre-training is $L_{\text{train}} = 512$.

| Wavelet type | Sequence Length | | | | | |
|---|---|---|---|---|---|---|
| | 128 | 256 | 512 | 1012 | 1512 | 2512 |
| Continuous Wavelet Families | | | | | | |
| Ricker | **23.64** | **20.82** | **19.19** | **18.23** | **18.00** | **17.99** |
| Gaussian | 23.77 | 20.90 | 19.30 | 18.31 | 18.02 | 17.88 |
| Morlet | 24.15 | 21.28 | 19.65 | 19.02 | 20.46 | 26.56 |
| Discrete Wavelet Families | | | | | | |
| Haar | 23.73 | 20.89 | 19.27 | 18.34 | 18.11 | 18.17 |
| db2 | 25.22 | 22.26 | 20.64 | 30.30 | 60.27 | 130.93 |
| db4 | 25.22 | 22.47 | 21.37 | 41.78 | 51.75 | 56.18 |
| db8 | 25.19 | 22.48 | 21.58 | 26.90 | 31.55 | 39.75 |
| db16 | 25.23 | 22.43 | 21.24 | 21.15 | 22.16 | 46.65 |
| db32 | 25.12 | 22.35 | 21.14 | 21.20 | 22.40 | 38.00 |
| sym2 | 25.11 | 22.21 | 20.68 | 31.25 | 61.00 | 126.32 |
| sym4 | 25.27 | 22.56 | 21.98 | 24.70 | 26.81 | 42.81 |
| sym8 | 29.27 | 26.13 | 24.63 | 23.97 | 31.47 | 92.36 |
| coif1 | 31.24 | 28.00 | 26.24 | 64.62 | 71.06 | 97.60 |
| coif2 | 25.24 | 22.47 | 21.39 | 27.74 | 27.39 | 44.26 |
| coif4 | 49.91 | 45.15 | 42.42 | 41.07 | 56.08 | 110.27 |
| coif8 | 25.15 | 22.39 | 21.26 | 21.31 | 22.26 | 35.73 |
| coif16 | 126.38 | 117.88 | 113.42 | 132.14 | 166.77 | 230.95 |
| dmey | 30.38 | 27.12 | 25.45 | 25.88 | 46.35 | 131.48 |
| bior1.3 | 26.27 | 23.36 | 23.69 | 23.38 | 30.71 | 88.66 |
| bior2.2 | 25.28 | 22.51 | 21.59 | 29.71 | 29.43 | 50.25 |
| bior2.6 | 25.29 | 22.70 | 21.60 | 22.15 | 22.71 | 40.61 |
| bior3.1 | 26.92 | 24.02 | 22.38 | 59.30 | 113.81 | 205.54 |
| bior3.5 | 25.17 | 22.49 | 21.65 | 27.41 | 27.19 | 53.99 |
| bior3.9 | 25.24 | 22.48 | 21.51 | 21.89 | 23.86 | 50.14 |
| bior4.4 | 25.52 | 22.72 | 21.64 | 21.67 | 24.42 | 51.46 |
| bior5.5 | 25.21 | 22.55 | 21.72 | 23.43 | 24.68 | 36.30 |
| bior6.8 | 25.14 | 22.39 | 21.21 | 21.10 | 22.31 | 46.97 |
| rbio1.1 | 24.26 | 21.34 | 19.69 | 18.79 | 18.63 | 18.98 |
| rbio1.3 | 25.28 | 22.50 | 21.39 | 52.06 | 47.78 | 59.94 |
| rbio2.2 | 25.92 | 23.08 | 21.98 | 68.57 | 86.12 | 93.90 |
| rbio2.6 | 25.29 | 22.68 | 21.60 | 24.54 | 24.47 | 44.57 |

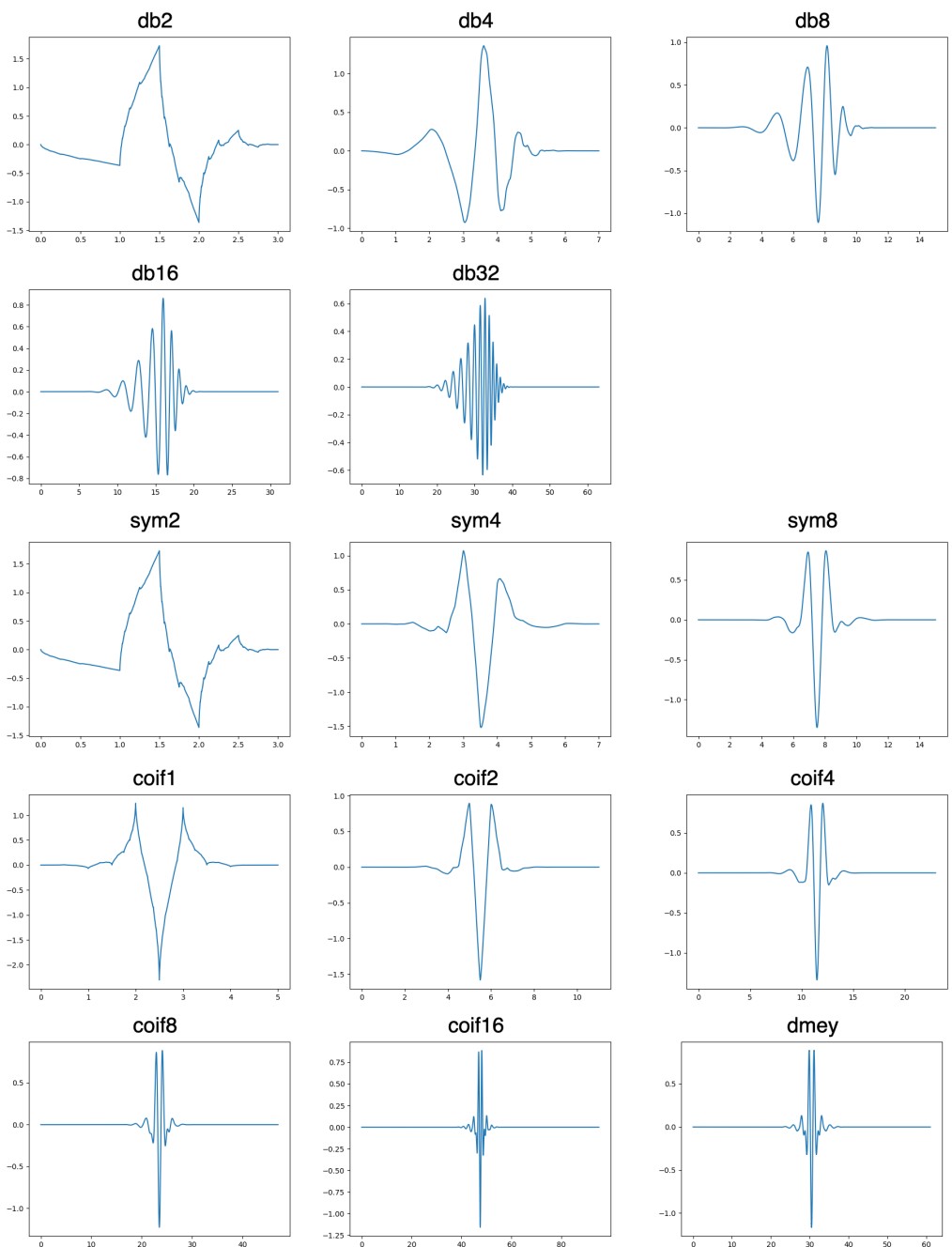

Figure 9: Graph of compared wavelets with level=10. Pywavelet (Lee et al., 2019) was used to calculate wavelets.

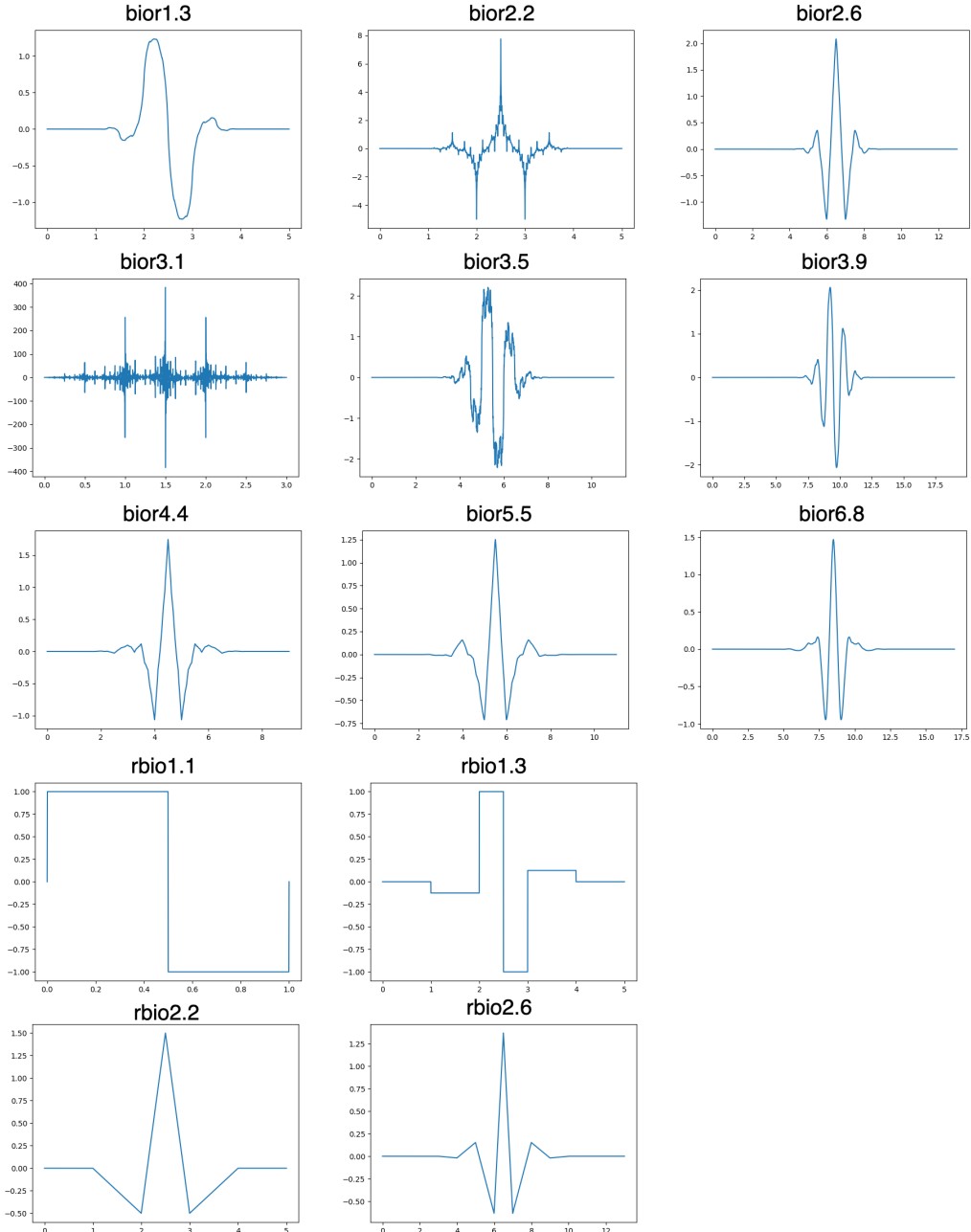

Figure 10: Graph of compared wavelets with level=10. Pywavelet (Lee et al., 2019) was used to calculate wavelets.

### A.15 EVALUATION ON LONGBENCH

The models pre-trained in Section 7 were evaluated on LongBench (Bai et al., 2024). This evaluation was conducted using a dataset that contained relatively long sentences. Furthermore, the multi-document QA task and single-document QA task were evaluated on all datasets. Since pre-training was conducted using an English dataset, evaluation was conducted using only the English dataset.

The results are shown in Figure 11. In some datasets, the performance of the model that adopted RoPE was good (Qasper, MuSiQue, and QMSum). In NarrativeQA, the two models attained almost the same score. However, in the remaining tasks, the proposed method was more effective. Note that this is an evaluation of a model that was pre-trained on a small dataset (redpajama-1B). As future work, it will be necessary to pre-train the model with a larger dataset and conduct evaluations with other models that are effective for long sentences, such as LongRangeArena (Tay et al., 2021).

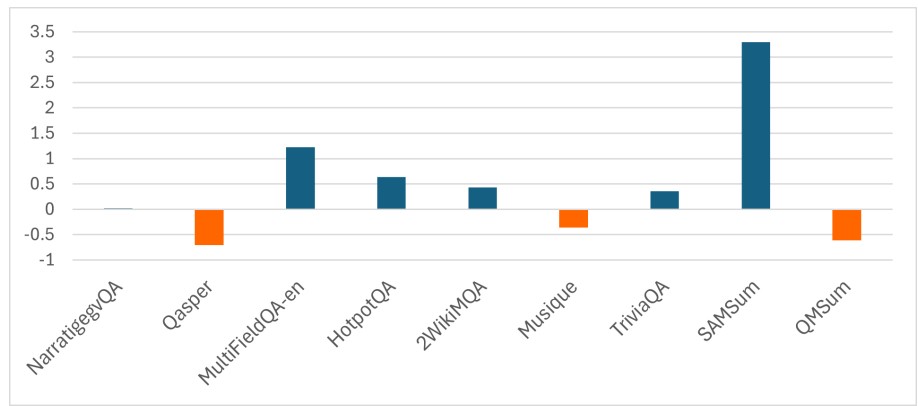

Figure 11: Evaluation results using LongBench(Bai et al., 2024). We evaluated the model pretrained in Section 7. The scores of the difference between the model using the proposed method, which uses wavelet-based position representation, and the model using RoPE are shown. The tasks were evaluated using the dataset, which contains relatively long sentences.

Table 5: Overview of the dataset statistics in LongBench (Bai et al., 2024). Avg len (average length) is computed using the number of words in the English.

| Dataset | Avg len | Metric | Samples |
|---|---|---|---|
| NarrativeQA | 18,409 | F1 | 200 |
| Qasper | 3,619 | F1 | 200 |
| MultiFieldQA-en | 4,559 | F1 | 150 |
| HotpotQA | 9,151 | F1 | 200 |
| 2WikiMQA | 4,887 | F1 | 200 |
| MuSiQue | 11,214 | F1 | 200 |
| TriviaQA | 8,209 | F1 | 200 |
| SAMSum | 6258 | Rouge-L | 200 |
| QMSum | 10614 | Rouge-L | 200 |

