# OpenReview forum: "Wavelet-based Positional Representation for Long Context"
_ICLR.cc/2025/Conference — ICLR 2025 Poster_

### Official Review · Reviewer_kKUx · 2024-11-02

**Soundness:** 2
**Presentation:** 2
**Contribution:** 2
**Rating:** 5
**Confidence:** 3

**Summary:**

This paper conducts a unified analysis of positional encoding methods in large language models and proposes a novel wavelet-based approach. The work begins by demonstrating that RoPE can be interpreted as a restricted wavelet transform using Haar-like filters operating at a fixed scale. The authors then analyze ALiBi, revealing that it functions similarly to windowed attention with varying window sizes but is limited by constraints on the attention mechanism's receptive field. Building on these insights, the paper introduces a new wavelet-based positional representation method that leverages multiple scales through wavelet transforms. The method is designed to capture both local and long-range dependencies without restricting the model's receptive field. The authors validate their approach through experiments on short and long contexts.

**Strengths:**

1. The motivation and connection are clear as the authors provide a unified analysis of the positional encoding methods.
2. The paper introduces a new wavelet-based positional representation method that leverages multiple scales through wavelet functions.
3. The proposed wavelet positional embedding shows improvements across tasks.

**Weaknesses:**

1. Novelty Concerns:
- The core idea of wavelet-based positional encoding has been explored in the previous work GMT (Ngo et al., 2023a,b), which uses graph wavelets to generate node positional representations that can capture the structural information of a center node on the graph at different resolutions, though in a different domain. However, there is insufficient discussion of how this approach differs fundamentally from or improves upon prior works.
2. Mathematical Rigor:
- The unified comparison for different positional encodings is not rigorous as the wavelet property needs stronger mathematical justification. The proposed "Haar-like wavelets" in Equation (7) require verification of wavelet admissibility conditions beyond just square integrability.
3. Experimental Design:
- Limited exploration of wavelet families (only four tested).
- Insufficient analysis of scale parameter choices (only two tested).
4. Minor issues:
- The paper lacks of clear notation definition and problem setup affects readability.
- Inconsistent terminology between abstract ("simple Haar wavelets" in Line 18) and technical content ("Haar-like wavelets" in Sec 3.2).
- The phrase 'single window size' (in Line 19) should be replaced with 'fixed scale parameter' to align with standard wavelet terminology, as it specifically refers to the dilation factor in wavelet transforms.
- The space definition L⊭(R) in Line 135 appears to have a typographical error - it should be L²(R), the space of square-integrable functions.
- Some mathematical notation is not explicitly defined. For example, the superscript T in equation (3) seems to represent the length of a discrete sequence, but this isn't explicitly defined.

**Questions:**

1. How does your approach fundamentally differ from MGT's wavelet position encoding?
2. Can you provide formal verification of the wavelet admissibility conditions for the proposed "Haar-like wavelets"? Or what else condition you might need to add to make it a wavelet?
3. Why were these specific wavelet families chosen? Have you explored other wavelet families, particularly discrete wavelets like Daubechies or biorthogonal wavelets?
4. What guided your choice of scale parameter ranges? Have you conducted sensitivity analyses on these choices?
5. What is the theoretical justification for removing the $\frac{1}{\sqrt{a}}$ amplitude term?

[1] Ngo, Nhat Khang, Truong Son Hy, and Risi Kondor. "Multiresolution graph transformers and wavelet positional encoding for learning long-range and hierarchical structures." The Journal of Chemical Physics 159.3 (2023).

---

> ### Author Response · Authors · 2024-11-23
> **Official Comment by Authors (1/3)**
>
> We would like to express our heartfelt gratitude to the reviewers for their valuable feedback in evaluating our research. Thank you for your excellent comments, especially regarding wavelet transforms. We would be more than happy to address your comments and provide our responses.
>
> > **(Q1) How does your approach fundamentally differ from MGT's wavelet position encoding?**
>
> This paper is of course included in the references. First of all, we would like to emphasize that we were quite worried about whether or not to mention this paper in the main text.
> Although this paper appears to be quite similar to our method, it is actually quite different.
>
> 1. **Position Encoding Models**: The models addressing position encoding are entirely different. We approach position encoding within the general Transformer framework, while the paper focuses on position encoding specific to Graph Transformers.
> 2. **Different Tasks**: The tasks being addressed are distinct. Their work involves tasks that handle graphs and represent macromolecules with multiple edges, whereas we are focused on sequences and extrapolation tasks.
> 3. **Divergent Objectives**: Their objective is to propose a new position encoding that guarantees locality in both spectral and spatial domains. In contrast, we propose a position encoding that remains valid outside of the learned context.
> 4. **Representation of Location**: The information regarding location representation diverges as well. They represent the position of each node in a graph, while we represent relative locations within a sequence. Graphs can have multiple edges, while sequences represent one-way paths, leading to fundamentally different ways to depict location.
>
> Despite the similar name and concept, our methods are quite distinct. We chose to include this paper in the references to avoid confusing readers by introducing it in the main text.
>
> ---
>
> > **(Q3)Why were these specific wavelet families chosen? Have you explored other wavelet families, particularly discrete wavelets like Daubechies or biorthogonal wavelets?**
>
> Thank you for your excellent point! We have conducted additional verification of other wavelets. The results of the experiment are described in Appendix A.11.
>
> We experimented with more than **20 additional wavelet types**.
> At this juncture, we are not incorporating wavelets that involve complex numbers. Should we decide to investigate the application of complex numbers to position encoding in the future, we anticipate that a revised strategy would be necessary, as referenced in [1].
> We found that the discrete wavelet approach did not yield the expected results; while we employed an approximation, we believe that a more effective selection strategy is needed.
> We also conducted a survey based on the vanishing moment and found that it may have a certain impact. This discovery would not have been possible without your insightful comments, for which we are sincerely grateful. Thank you very much!
> Our primary goal in this study was to establish foundational principles of position encoding using wavelet transforms, so we consider the discrete wavelet approach an area for future exploration.
>
> ---
>
> > **(Q4) What guided your choice of scale parameter ranges? Have you conducted sensitivity analyses on these choices?**
>
> In response to your suggestions, we have included an ablation study examining the impact of shift and scale parameters, presented in Appendix A.10. We experimented with more than **10 additional parameter patterns**.
>
> The following was learned from the test results. Increasing the scale parameter while keeping the shift parameter constant generally maintained extrapolation performance, though with some fluctuations. However, increasing the number of shift parameters while decreasing scale parameters led to a decline in performance, highlighting the importance of scale parameters. Conversely, adding more scale parameters while reducing shift parameters improved performance in some cases. Nevertheless, reducing shift parameters to two or none resulted in worse extrapolation performance, indicating that shift parameters are also significant.

---

> ### Author Response · Authors · 2024-11-23
> **Official Comment by Authors (2/3)**
>
> > **(Q5) What is the theoretical justification for removing the  amplitude term?**
>
> This is a kind of normalization to make the effects of positional expressions even. This is not a theory, but an implementation technique. In order to make the loss converge, it was necessary to make the amplitude of the wavelet values between -1 and +1. If the amplitude of the wavelet was increased, the loss did not converge.
>
> ---
> > **(Q2) Can you provide formal verification of the wavelet admissibility conditions for the proposed "Haar-like wavelets"? Or what else condition you might need to add to make it a wavelet?**
>
> > **(Weakness)  Minor issues**
>
> We are currently conducting a thorough review of Section 3.2. We sincerely apologize for keeping you waiting for such a long time and kindly ask for your patience a little longer. We will respectfully share the modified text with you within the rebuttal period.
>
> [1] Wang+, ICLR2020. Encoding word order in complex embeddings

---

> ### Author Response · Authors · 2024-11-27
> **Official Comment by Authors (3/3)**
>
> We sincerely appreciate the reviewers for their time, effort, and valuable feedback in evaluating our research.
> We have completed the additional proof and revisions for the points you kindly pointed out.
>
> > **(Weakness) Minor issues: Inconsistent terminology between abstract ("simple Haar wavelets" in Line 18) and technical content ("Haar-like wavelets" in Sec 3.2).**
>
> > **(Weakness) Minor issues: The phrase 'single window size' (in Line 19) should be replaced with 'fixed scale parameter' to align with standard wavelet terminology, as it specifically refers to the dilation factor in wavelet transforms.**
>
> > **(Weakness) Minor issues: The space definition L⊭(R) in Line 135 appears to have a typographical error - it should be L²(R), the space of square-integrable functions.**
>
> > **(Weakness) Minor issues: Some mathematical notation is not explicitly defined. For example, the superscript T in equation (3) seems to represent the length of a discrete sequence, but this isn't explicitly defined.**
>
> Thank you very much for pointing this out. We have corrected the terms you pointed out and the typo.
>
> > **(Q2) Can you provide formal verification of the wavelet admissibility conditions for the proposed "Haar-like wavelets"? Or what else condition you might need to add to make it a wavelet?**
>
> Thank you for your wonderful point! We have revisited and revised the definitions of terms in Section 3.2 to ensure a clearer understanding (text in red).  Additionally, we have reconsidered the conditions for $\psi (t)$ and $\psi (t)$ in Eq.(7) to be wavelets and have corrected them to include the previously missing zero-mean property (in Appendix A15).
> Furthermore, we have added a proof in the appendix A15 to demonstrate the existence of $f(t)$ and $\delta (t)$ that satisfy these conditions.
>
> ----
>
> Furthermore, it was found that there was an error in the experiment in Section 7, so it was re-evaluated. (The length that should have been reported was incorrect.) As a result of the re-evaluation, it was confirmed again that it was more effective than RoPE. The scores in Table 2 have been updated.
>
> We sincerely apologize for the delay in reporting the results of the additional experiments. We also deeply appreciate your extremely valuable comments. If you have any questions or require further clarification, please do not hesitate to let us know. We look forward to hearing from you.

---

### Official Review · Reviewer_DFcs · 2024-11-03

**Soundness:** 3
**Presentation:** 3
**Contribution:** 3
**Rating:** 6
**Confidence:** 4

**Summary:**

The paper proposes a wavelet transform-based positional representation for transformer models. It begins with highlighting the properties of existing position representation techniques such as relative position bias, RoPE and ALiBi. The authors then show how RoPE can be interpreted as a single scale wavelet transform with a Haar-like wavelet. Then the authors present a position embedding based on wavelet transform. This multi-scale embedding can be viewed as a generalization of RoPE and possess attractive "multi-window" properties of ALiBi. Experiment results have been reported on short and longer-context scenarios showing improved perplexity over existing methods and better extrapolation properties.

**Strengths:**

- The paper is well-written barring some typographical errors. It motivates the problem well, starts with a well-defined goal, describes existing methods clearly, and presents the proposed method in a manner which is easy to appreciate.
- The paper tackles a critical problem of context length extrapolation which often arises in practical settings. The method holds significance, not just for the language modeling community, but also other domains such as time series forecasting where context length extrapolation may improve the performance of models on high-frequency data (see [1]).
- The proposed method is novel to the best of my knowledge. It creatively relates RoPE to time-frequency analysis and proposes a promising method based on wavelet transforms.
- The method outperforms alternatives and scales gracefully with extrapolated context lengths.

[1] Ansari, Abdul Fatir, et al. "Chronos: Learning the language of time series." arXiv preprint arXiv:2403.07815 (2024).

**Weaknesses:**

- While the discussion of related methods is generally well done, discussion of RoPE scaling techniques (linear, NTK-aware) is missing. A discussion and comparison with these techniques would significantly improve the positioning of this work.
- The results reported in sections 6 and 7 are excellent proofs of concept but they lack comprehensiveness. Particularly, in section 7, evaluations beyond the CodeParrot dataset would be needed to thoroughly appreciate the proposed method. Furthermore, the paper only evaluates the model in term of the perplexity. For a stronger evaluation, this needs to be augmented with task-based evaluations. Apart from common language tasks, you might also want to consider associative recall tasks [2] and the long range arena [3] since the focus of this work is context length extrapolation. To further demonstrate the robustness of the proposed method, domains beyond natural language may also be considered, for example, DNA modeling [4], audio generation [4] and time series forecasting [1].

I am willing to raise my score if the evaluation is strengthened.

[1] Ansari, Abdul Fatir, et al. "Chronos: Learning the language of time series." arXiv preprint arXiv:2403.07815 (2024).
[2] Arora, Simran, et al. "Zoology: Measuring and improving recall in efficient language models." arXiv preprint arXiv:2312.04927 (2023).
[3] Tay, Yi, et al. "Long range arena: A benchmark for efficient transformers." arXiv preprint arXiv:2011.04006 (2020).
[4] Gu, Albert, and Tri Dao. "Mamba: Linear-time sequence modeling with selective state spaces." arXiv preprint arXiv:2312.00752 (2023).

**Questions:**

See above.

---

> ### Author Response · Authors · 2024-11-23
> **Official Comment by Authors (1/2)**
>
> We would like to express our heartfelt gratitude to the reviewers for their valuable feedback in evaluating our research. We would also like to sincerely thank you for the relatively high evaluation we have received. We are currently conducting additional experiments in response to the comments provided and will respectfully share the results during the rebuttal period.
>
> Although this is not a response to the comments we received, we have included the following points in the appendix.
> - Additional investigation of shift and scale parameters (A.9)
> - Additional investigation of other wavelet types (A.10)
> - Further explanation of the attention map (A.11)
>
> There were also some errors in the mathematical formulae, which I have corrected to the extent possible at this stage. We are also currently conducting a thorough review of Section 3.2. We sincerely apologize for keeping you waiting for such a long time and kindly ask for your patience a little longer.

---

> ### Author Response · Authors · 2024-11-27
> **Official Comment by Authors (2/2)**
>
> We sincerely appreciate the reviewers for their time, effort, and valuable feedback in evaluating our research. We have completed additional experiments and would like to share the results.
> In addition, we have added a discussion on the method of position interpolation.
>
> > **(Weakness 1) While the discussion of related methods is generally well done, discussion of RoPE scaling techniques (linear, NTK-aware) is missing. A discussion and comparison with these techniques would significantly improve the positioning of this work.**
>
> In most cases, the method of position interpolation has been verified using large-scale language models such as llama. **Therefore, we have added a discussion to Section 7.2 (text in red), which contains experiments using llama.**
> To summarize, we believe that the position interpolation method such as NTK, PI, YaRN and LongRoPE can be incorporated into our wavelet-based method. Both $\theta$ in RoPE and the scale parameter in our method represent the upper limit of the position representation. Therefore, we believe that $\theta$'s position interpolation can also be used for interpolating our scale parameter. We will consider this verification as a future issue.
> Furthermore, the LongRoPE paper [Ding+, Arxiv2024] reports that performance can be improved by avoiding the first position. We think this feature is similar to our shift parameter, and we think that methods like LongRoPE can also be applied to our method.
>
> >(**Weakness 2) The results reported in sections 6 and 7 are excellent proofs of concept but they lack comprehensiveness. Particularly, in section 7, evaluations beyond the CodeParrot dataset would be needed to thoroughly appreciate the proposed method. Furthermore, the paper only evaluates the model in term of the perplexity. For a stronger evaluation, this needs to be augmented with task-based evaluations. Apart from common language tasks, you might also want to consider associative recall tasks [2] and the long range arena [3] since the focus of this work is context length extrapolation. To further demonstrate the robustness of the proposed method, domains beyond natural language may also be considered, for example, DNA modeling [4], audio generation [4] and time series forecasting [1].**
>
> We conducted additional experiments based on the comments we received. Initially, we considered conducting experiments on the Long Range Arena[Tay+, ICLR2021], but due to time and computing resource constraints, we were unable to do so. **Instead, we conducted additional verification on the LongBench task. The details of the experiment are described in Appendix A14.** We conducted verification on the LongBench task, which has relatively long contexts. We used the following datasets: NarrativeQA, Qasper, MultiFieldQA-en, HotpotQA, 2WikiMQA, MuSiQue, TriviaQA, SAMSum, and QMsum, and evaluated them using F1 score or Rouge-L. The experimental results showed that it was more effective than RoPE for most tasks.
>
> ---
>
> Furthermore, it was found that there was an error in the experiment in Section 7, so it was re-evaluated. (The length that should have been reported was incorrect.) As a result of the re-evaluation, it was confirmed again that it was more effective than RoPE. The scores in Table 2 have been updated.
>
> We sincerely apologize for the delay in reporting the results of the additional experiments. We also deeply appreciate your extremely valuable comments.
> If you have any questions or require further clarification, please do not hesitate to let us know.
> We look forward to hearing from you.
>
> [Ding+, Arxiv2024] LongRoPE: Extending LLM Context Window Beyond 2 Million Tokens
>
> [Tay+, ICLR2021] Long Range Arena : A Benchmark for Efficient Transformers ,ICLR 2021

---

### Official Review · Reviewer_fCa9 · 2024-11-04

**Soundness:** 2
**Presentation:** 2
**Contribution:** 2
**Rating:** 5
**Confidence:** 4

**Summary:**

the paper presents a novel positional encoding method leveraging wavelet transforms to address the challenges of limited receptive field and  extrapolation in LLMs.

The proposed wavelet-based approach aims to overcome these limitations by introducing a multi-scale analysis that captures the fluidity of natural language and enhances the model's ability to extrapolate beyond its training context length.

In extrapolation experiments, model that used Ricker-based wavelet positional embedding had the lowest PPL trained on the WikiText-103 dataset, compared with RoPE, ALibi, Transformer-XL.

**Strengths:**

- The authors provide a solid theoretical foundation by drawing parallels between RoPE and wavelet transforms, and by extending this analogy to propose their method.
- The proposed method has the potential to be widely applicable to various transformer-based models
- The paper is easy to read and positions itself clearly with respect to related work.

**Weaknesses:**

- The paper primarily evaluates the method on language modeling tasks. It would be valuable to see how the approach generalizes to other NLP tasks such as question answering or text summarization.
- The paper does not provide sufficient experimental evidence to support the authors' claim that Wavelet Transform can capture the dynamic changes in a sequence over positions.（L84-85）
- See my questions/suggestions below

**Questions:**

- How did the “shift and scale parameters” been decided? Is there any ablation study or results for this particular setting?
- Did you compare the length extrapolation results with other RoPE-based methods? (eg. relative works you mentioned in L47-L48)
- Both RoPE and Wavelet methods aim to model distances at different scales by placing them in different dimensions [1]. In Figure 3, the diagonal lines in the RoPE plot appear because the model can focus on the features of specific token distances. However, Wavelet functions do not have this stable frequency inductive bias, which is why they do not produce diagonal patterns in the attention scores. Do you think this is a desirable property?
- In the experiment represented by Figure 3, what kind of text is input into the model? Why is the initial word particularly important? Which specific tokens does the model attend to? Is this phenomenon common?

[1] Hong, Xiangyu, et al. "On the token distance modeling ability of higher RoPE attention dimension." arXiv preprint arXiv:2410.08703 (2024).

---

> ### Author Response · Authors · 2024-11-23
> **Official Comment by Authors (1/2)**
>
> We sincerely appreciate the reviewers for their time, effort, and valuable feedback in evaluating our research. We are also deeply grateful for your thoughtful assessment of the readability and the theoretical derivations presented in our paper. In response to the reviewers' valuable comments, we have conducted additional explanations and experiments, and we would like to respectfully present our findings for your consideration.
>
> > **(Q) How did the “shift and scale parameters” been decided? Is there any ablation study or results for this particular setting?**
>
> In response to your suggestions, we have included an ablation study examining the impact of shift and scale parameters, presented in Appendix A.10, as well as an analysis of each wavelet type in Appendix A.11. **In the parameter abration study, we verified 10 patterns of parameters. In the wavelet-type abration study, we used more than 20 wavelets.** Our findings reveal that adjusting the shift and scale parameters leads to a further reduction in perplexity. However, we found that the discrete wavelet approach did not yield the expected results; while we employed an approximation, we believe that a more effective selection strategy is needed.
>
> ---
>
> > **(Q) Both RoPE and Wavelet methods aim to model distances at different scales by placing them in different dimensions [1]. In Figure 3, the diagonal lines in the RoPE plot appear because the model can focus on the features of specific token distances. However, Wavelet functions do not have this stable frequency inductive bias, which is why they do not produce diagonal patterns in the attention scores. Do you think this is a desirable property?**
>
> We appreciate the opportunity to address the concerns regarding the presence of diagonal patterns in the attention scores. We firmly believe that the absence of such patterns is a favorable attribute of RoPE. Figure 3 show the attention score in extrapolation. The length of the training is 512 and the inference length is 1012. Notably, Diagonal patterns appear in places longer than 512. When it is shorter than 512, diagonal patterns like this do not exist.This observation leads us to conclude that if Rotary Position Embedding (RoPE) is capable of effectively recognizing positions, diagonal patterns should not manifest. While we acknowledge the importance of further investigating the presence of diagonal patterns, we consider this aspect to be outside the primary focus of our current study. Therefore, we will not be conducting additional verification at this time, but we welcome future exploration of this topic. Thank you for your understanding.
>
> ---
>
> > **(Q) In the experiment represented by Figure 3, what kind of text is input into the model? Why is the initial word particularly important? Which specific tokens does the model attend to? Is this phenomenon common?**
>
> **An example of the correspondence between the text and the heat map is shown in Appendix A.13.** The model paid particular attention to special tokens such as <s>. In addition, for some heads, it also paid attention to words corresponding to the subject, which may have captured the characteristics of the sentence. This phenomenon was observed in any text.
>
> We are currently conducting additional experiments in response to the other comments provided. We will respectfully share the results with you within the rebuttal period.

---

> ### Author Response · Authors · 2024-11-27
> **Official Comment by Authors (2/2)**
>
> We sincerely appreciate the reviewers for their time, effort, and valuable feedback in evaluating our research. We will reply to your next comment.
>
> >**(Q) Did you compare the length extrapolation results with other RoPE-based methods? (eg. relative works you mentioned in L47-L48)**
>
> Although these position interpolation methods need to be discussed, we believe that they are outside the scope of our paper.
> NTK, PI, and YaRN are methods that approach RoPE's $theta$, but they require fine-tuning with longer contexts after pre-training. In this paper, we are focusing on position encoding during pre-training, so position interpolation through fine-tuning is not covered.
> The same applies to LongRoPE. They are not covered because they perform parameter optimization as well as fine-tuning.
>
> **On the other hand, we think that a discussion related to these position interpolations is necessary, so we have added a discussion to Section 7.2 (text in red).**
>
>
> ---
> > **(Weakness) The paper primarily evaluates the method on language modeling tasks. It would be valuable to see how the approach generalizes to other NLP tasks such as question answering or text summarization.**
>
> We have also added the results of the LongBench[7] evaluation in appendix A.14.
> LongBench includes summary and QA tasks, and is evaluated using F1 and Rouge scores. We evaluated  the QA and summary tasks, which have relatively long sequences.
> We used the following datasets: NarrativeQA, Qasper, MultiFieldQA-en, HotpotQA, 2WikiMQA, MuSiQue, TriviaQA, SAMSum, and QMsum, and evaluated them using F1 score or Rouge-L.
> The results show that the proposed method outperforms RoPE in most cases.
>
> ---
> > **(Weakness) The paper does not provide sufficient experimental evidence to support the authors' claim that Wavelet Transform can capture the dynamic changes in a sequence over positions.（L84-85）**
>
> The assertion that wavelet transforms can capture dynamic changes in sequences is not specific to our method but rather a general characteristic of wavelet transforms themselves. This feature has been discussed in considerable detail in the book of wavelet [1], and since it is a well-established property, we omit the proof of wavelet transforms' ability to capture dynamic changes in this paper.
>
> ---
>
> Furthermore, it was found that there was an error in the experiment in Section 7, so it was re-evaluated. (The length that should have been reported was incorrect.) As a result of the re-evaluation, it was confirmed again that it was more effective than RoPE. The scores in Table 2 have been updated.
>
> We also deeply appreciate your extremely valuable comments.
>
> If you have any questions or require further clarification, please do not hesitate to let us know.
> We look forward to hearing from you.
>
> [1] Ingrid Daubechies. Ten Lectures on Wavelets

---

### Official Review · Reviewer_QwSG · 2024-11-10

**Soundness:** 2
**Presentation:** 2
**Contribution:** 2
**Rating:** 5
**Confidence:** 4

**Summary:**

In this work, the authors proposed wavelet-based positional encoding for the length extrapolation problem of language models. The motivation is based on an observation that the widely used Rope is related to wavelet transform via simple Haar wavelet functions with a fixed scale. By further investigating the properties of ALiBi, the authors modified the form of relative positional encoding to use wavelet transform based approaches. Several experiments are conducted to demonstrate the performance of the proposed positional encoding.

**Strengths:**

1. The paper is easy-to-follow.
2. The length extrapolation problem is important for language models.

**Weaknesses:**

1. **There exist gaps from the motivation to the proposed approach**. In Section 3 and 4, the authors provide analysis on the relationship between RoPE and wavelet transform, and the properties of ALiBi like positional encodings. The ability of ALiBi to accommodate multiple window sizes is concluded as the key point for better length extrapolation performance, while RoPE is claimed to be worse. Based on these statements, a natural question is, what is the advantage of using wavelet transform? The authors do not provide enough supportive quantitative evidence for it, but directly modify the relative positional encoding by using wavelet transform. If it has superiorities, why don't we follow the form of RoPE to use wavelet transform instead of RPE? These points should be better clarified to make the motivation of the proposed approach more convincing.

2. **The design choice for the proposed approach should be better supported**. In Section 5, the authors propose to use Ricker Wavelet as the base function of wavelet transform, and set the shift and amplititude parameters with predefined strategies. However, the reasons to choose these designs are not well clarified (except lines 294-303, which are rather superficial).

3. **The empirical improvement is marginal**. From Table 2 and 3, we can see that the proposed PE does not show significant improvement compared to compared approaches such as ALiBi, which is proposed in 2022 and is not the state-of-the-art nowadays. The evaluations are also limited in task types and scales. Overall, the empirical results are not convincing to show the superiority of the proposed approach.

Overall, I hope the authors can well address the above concerns, which I think are important for the quality of this work.

**Questions:**

See the Weaknesses.

---

> ### Author Response · Authors · 2024-11-23
> **Official Comment by Authors (1/3)**
>
> We sincerely appreciate the reviewers for their time, effort, and valuable feedback in evaluating our research.  And, we would also like to thank you for evaluating the readability of our paper. In response to the reviewers' valuable comments, we have conducted additional explanations and experiments, and we would like to respectfully present our findings for your consideration.
>
> > **(Weakness1) There exist gaps from the motivation to the proposed approach. In Section 3 and 4, the authors provide analysis on the relationship between RoPE and wavelet transform, and the properties of ALiBi like positional encodings. The ability of ALiBi to accommodate multiple window sizes is concluded as the key point for better length extrapolation performance, while RoPE is claimed to be worse. Based on these statements, a natural question is, what is the advantage of using wavelet transform? The authors do not provide enough supportive quantitative evidence for it, but directly modify the relative positional encoding by using wavelet transform. If it has superiorities, why don't we follow the form of RoPE to use wavelet transform instead of RPE? These points should be better clarified to make the motivation of the proposed approach more convincing.**
>
> Thank you for presenting such a thought-provoking question!
>
> > Based on these statements, a natural question is, what is the advantage of using wavelet transform?
>
> Initially, we contemplated the application of wavelet transform in accordance with the RoPE format. However, as outlined in Section 3.2, "Theoretical Analysis," it is important to note that RoPE conducts wavelet transform along the dimensional axis rather than the time axis. This approach with RoPE does not fully use the unique characteristics of wavelet transforms, which investigate the information in a signal at a certain time.
>
> Furthermore, when applying wavelet transformation based on RoPE, there is the issue that the scale parameter can only be set to a value less than or equal to d_head. When applying wavelet transformation based on RPE, the scale parameter can be set to any value within the context length when training. In wavelet transformation, the scale parameter represents resolution, and this characteristic is very important (we believe that the importance of scale can be understood from the ALiBi validation section and the ablation study for each scale parameter that we have added this time. ).
>
> Additionally, even if positional encoding were to incorporate a wavelet transform based on RoPE, the reliance on absolute position could hinder any potential improvements in extrapolation performance. To truly capitalize on the benefits of wavelet transforms for extrapolation purposes, it is essential that positional encoding shifts focus to performing the wavelet transform along the time axis or position, without adopting absolute position. This adjustment would allow for a more effective utilization of wavelet characteristics.
>
> ---
>
> > **(Weakness2) The design choice for the proposed approach should be better supported. In Section 5, the authors propose to use Ricker Wavelet as the base function of wavelet transform, and set the shift and amplititude parameters with predefined strategies. However, the reasons to choose these designs are not well clarified (except lines 294-303, which are rather superficial).**
>
> In response to your suggestions, we have included an ablation study examining the impact of shift and scale parameters, presented in Appendix A.10, as well as an analysis of each wavelet type in Appendix A.11. **In the parameter abration study, we verified 10 patterns of parameters. In the wavelet-type abration study, we used more than 20 wavelets.** Our findings reveal that adjusting the shift and scale parameters leads to a further reduction in perplexity. However, we found that the discrete wavelet approach did not yield the expected results; while we employed an approximation, we believe that a more effective selection strategy is needed. Our primary goal in this study was to establish foundational principles of position encoding using wavelet transforms, so we consider the discrete wavelet approach an area for future exploration. We chose Ricker, Morlet, and Gaussian wavelets due to their status as the most representative wavelets commonly described by mathematical formulas. At this juncture, we are not incorporating wavelets that involve complex numbers. Should we decide to investigate the application of complex numbers to position encoding in the future, we anticipate that a revised strategy would be necessary, as referenced in [1].

---

> ### Author Response · Authors · 2024-11-23
> **Official Comment by Authors (2/3)**
>
> > **(Weakness 3)The empirical improvement is marginal. From Table 2 and 3, we can see that the proposed PE does not show significant improvement compared to compared approaches such as ALiBi, which is proposed in 2022 and is not the state-of-the-art nowadays. The evaluations are also limited in task types and scales. Overall, the empirical results are not convincing to show the superiority of the proposed approach.**
>
> I would like to discuss whether ALiBi is a method of SOTA.
> As noted in the introduction of Section 1, our research specifically investigates position encodings used during pre-training. Although ALiBi was published in 2022 [2], it is still considered to be state-of-the-art as a position encoding used in “pre-training” because it is also used in mpt models[3] that can handle long sentences with high-performance. Furthermore, our comparison includes NoPE and RoPE, with θ=500000, as indicated in Table 2. Notably, NoPE is a recent 2023 publication, while RoPE (with θ=0.5m) is featured in a 2024 paper. It's important to highlight that RoPE (θ=0.5m) is employed in the Llama-3 model, which has achieved numerous state-of-the-art benchmarks in various tasks. The majority of current large-scale language models utilize either RoPE or ALiBi. We propose that our method offers distinct advantages, as it effectively captures intermediate words (as detailed in Section 6.3.2 and illustrated in Figure 3) while maintaining strong extrapolation performance (as shown in Section 6.2, Table 2), thereby demonstrating clear superiority over these established position encodings.
>
>
> **In response to the comments we received, we have changed the notation of the proposed methods in Tables 1 and 2. (e.g., RoPE($θ$ =0.5m) -> RoPE (Xiong et al., 2024))**
>
> **We have also added the results of the LongBench[7] evaluation in appendix A.14. The results show that the proposed method outperforms RoPE in most cases.**
>
>
> Thank you for your useful comments. We are currently working on a comparison experiment between other tasks and the state-of-the-art RoPE position interpolation method. We will definitely let you know the results during the rebuttal period.
>
> ---
>
> [1] Wang+, ICLR2020. Encoding word order in complex embeddings
>
> [2] Press+, ICLR2022. Train Short, Test Long: Attention with Linear Biases Enables Input Length Extrapolation
>
> [3] MosaicML NLP Team, Online2023. Introducing MPT-7B: A New Standard for Open-Source, Commercially Usable LLMs
>
> [4] Kazemnejad+, NeurIPS 2023.The Impact of Positional Encoding on Length Generalization in Transformers
>
> [5] Xiong+, NAACL 2024. Effective Long-Context Scaling of Foundation Models
>
> [6] Dubey+, Arxiv 2024. The Llama 3 Herd of Models
>
> [7] Bai+, ACL2024. LongBench: A Bilingual, Multitask Benchmark for Long Context Understanding

---

> ### Author Response · Authors · 2024-11-27
> **Official Comment by Authors (3/3)**
>
> We sincerely appreciate the reviewers for their time, effort, and valuable feedback in evaluating our research.
> We would like to inform you that we have updated the paper in response to the following comments.
>
> >**(Weakness 1) There exist gaps from the motivation to the proposed approach. In Section 3 and 4, the authors provide analysis on the relationship between RoPE and wavelet transform, and the properties of ALiBi like positional encodings. The ability of ALiBi to accommodate multiple window sizes is concluded as the key point for better length extrapolation performance, while RoPE is claimed to be worse. Based on these statements, a natural question is, what is the advantage of using wavelet transform? ....**
>
> Considering the application of wavelet transformation to RoPE, the simplest formula would be the one in Appendix A13, Formula (25).
> (It's a fairly large formula, so it's impossible to include it here. Please refer to the paper.)
>
> **We implemented this approach; however, the computational cost was over five times higher than anticipated, and the pre-training did not complete.** Given the current landscape of large-scale language models, this cost is a significant concern. Additionally, there are key differences between RoPE-based wavelet transformation and RPE-based wavelet transformation, leading us to conclude that RPE-based wavelet transformation is more practical.
>
> The differences between RoPE-based Wavelet and RPE-based Wavelet are as follows:
> - **Number of Scale Parameters:** In RPE-based Wavelet, the scale parameters can be selected up to the maximum sequence length. However, in RoPE-based Wavelet, the selection is limited to a maximum of \( d \).
> - **Memory Usage:** RoPE-based Wavelet requires a wavelet matrix that corresponds to the number of absolute positions \( m \). Consequently, the memory usage is significantly higher. Unlike RoPE-based, RPE-based Wavelet does not necessitate a wavelet matrix that matches \( m \) values, allowing the use of Tip 2 from Appendix A4, which improves memory efficiency.
> - **Absolute and Relative Positions:** When applying wavelet transforms using RoPE-based, it is necessary to use absolute positions. In contrast, RPE-based can utilize relative positions, which enhances extrapolation.
> - **Computational Cost:** Implementing wavelet transforms via RoPE-based requires processing both the query and the key, necessitating two calculations. RPE-based Wavelet only requires one computation since it processes only the query.
>
> ---
>
>
> Furthermore, it was found that there was an error in the experiment in Section 7, so it was re-evaluated. (The length that should have been reported was incorrect.) As a result of the re-evaluation, it was confirmed again that it was more effective than RoPE. The scores in Table 2 have been updated.
>
> We also deeply appreciate your extremely valuable comments.
> If you have any questions or require further clarification, please do not hesitate to let us know. We look forward to hearing from you.

---

### Author Response · Authors · 2024-11-25
**Official Comment by Authors**

Dear reviewers,

Thank you for taking the time to review this paper. We have added experiments and notes in response to some of the comments, and have reported them. As the discussion period is now coming to an end, we would be grateful if you could discuss them.

We are also currently conducting experiments on state-of-the-art RoPE improvement methods and other tasks. We will report the results of these experiments as soon as they are ready.
We look forward to discussing them with you!

---

### Author Response · Authors · 2024-11-29

Dear Reviewers,

Thank you very much for taking the time to review our manuscript. We deeply appreciate your thoughtful comments and suggestions.
In response to the valuable feedback we received, we have revised our manuscript accordingly and conducted additional experiments to address the points raised. We have carefully documented these changes and believe that your insights have significantly improved the quality of our work. We are truly grateful for your constructive input.

If you have any additional comments or further suggestions, we would greatly appreciate hearing from you. We understand that you are very busy, and we sincerely apologize for any inconvenience caused. However, we would be most grateful if you could kindly provide any feedback at your earliest convenience to help facilitate the review process.
Thank you again for your time and effort. We greatly appreciate your guidance and look forward to hearing from you.

---

### Meta-Review · Area_Chair_M7PL · 2024-12-23

**Metareview:**

The paper proposes a wavelet transform-based positional representation for transformer models. This multi-scale embedding can be viewed as a generalization of RoPE and possess attractive "multi-window" properties of ALiBi. Several experiments are conducted in long and short context scenarios to demonstrate the performance of the proposed positional encoding. They show improved perplexity, better length extrapolation properties and  sometimes even improved question answering performance compared to RoPE.

Strengths: The paper tackles a critical problem of context length extrapolation which often arises in practical settings. The proposed method is novel, and the authors provide a solid theoretical foundation by drawing parallels between RoPE and wavelet transform. Their unified analysis of positional encoding methods sheds more  light on this important topic.

Weaknesses: Reviewers raised a few concerns regarding novelty, mathematical rigor and experimental evaluation.  In my estimation they have been adequately addressed by the authors during the discussion phase.

Overall I think this is a good paper which adds theoretical understanding, as well as a practical method with potential significance beyond the language learning community. I recommend accepting it.

**Additional Comments On Reviewer Discussion:**

The reviewers did not respond to the authors during discussion, so I had to judge the responses myself.
In response to the concerns regarding mathematical rigor the authors clarified several points and fixed a few shortcomings in their theoretical analysis.
Regarding experimental evaluation, the authors added several new evaluations and a host of additional ablations. Some concerns regarding comparison to additional baselines and larger models remain, but are not critical in my opinion.

---

### Decision · Program_Chairs · 2025-01-22

Accept (Poster)